# A chromosome-scale assembly reveals chromosomal aberrations and exchanges generating genetic diversity in *Coffea arabica* germplasm

Simone Scalabrin [1,10], Gabriele Magris [2,3,10], Mario Liva[1,2,3,10], Nicola Vitulo [4], Michele Vidotto[1], Davide Scaglione [1], Lorenzo Del Terra[5], Manuela Rosanna Ruosi[6], Luciano Navarini [5], Gloria Pellegrino[6], Jorge Carlos Berny Mier y Teran [7], Lucile Toniutti[7,8,9], Furio Suggi Liverani[5], Mario Cerutti[6], Gabriele Di Gaspero [2] ✉ & Michele Morgante [2,3] ✉

In order to better understand the mechanisms generating genetic diversity in the recent allotetraploid species *Coffea arabica*, here we present a chromosome-level assembly obtained with long read technology. Two genomic compartments with different structural and functional properties are identified in the two homoeologous genomes. The resequencing data from a large set of accessions reveals low intraspecific diversity in the center of origin of the species. Across a limited number of genomic regions, diversity increases in some cultivated genotypes to levels similar to those observed within one of the progenitor species, *Coffea canephora*, presumably as a consequence of introgressions deriving from the so-called Timor hybrid. It also reveals that, in addition to few, early-occurring exchanges between homoeologous chromosomes, there are numerous recent chromosomal aberrations including aneuploidies, deletions, duplications and exchanges. These events are still polymorphic in the germplasm and could represent a fundamental source of genetic variation in such a lowly variable species.

Long-read sequencing, based on either Pacific Biosciences (PacBio) or Oxford Nanopore Technologies (ONT)[1], has revolutionized the field of genome assembly[2–4], providing unprecedented opportunities to obtain contiguous and accurate chromosome sequences for complex genomes.

*Coffea arabica* is an allotetraploid hybrid of *C. eugenioides* and *C. canephora* (Robusta coffee) and contributes to approximately 60% of

world coffee production. It has an estimated genome size of 1.33 Gbp based on flow cytometry assays[5]. A number of partial assemblies are available for *C. arabica*. Whole-genome sequencing (WGS) of pooled BAC clones has been performed using Illumina short reads, leading to a genome assembly that provided high sequence accuracy and reliable separation of homoeologous regions but was affected by high fragmentation[6]. Later on, a WGS shotgun approach with low-coverage

[1]IGA Technology Services, 33100 Udine, Italy. [2]Istituto di Genomica Applicata, 33100 Udine, Italy. [3]Department of Agricultural, Food, Environmental and Animal Sciences, University of Udine, 33100 Udine, Italy. [4]Department of Biotechnology, University of Verona, 37134 Verona, Italy. [5]illycaffè SpA, 34147 Trieste, Italy. [6]Lavazza Group, 10152 Torino, Italy. [7]World Coffee Research, Portland 97225 OR, USA. [8]Present address: CIRAD, UMR AGAP Institut, 97130 Capesterre-Belle-Eau, Guadeloupe, France. [9]Present address: UMR AGAP Institut, University of Montpellier, CIRAD, INRAE, Institut Agro, 34060 Montpellier, France. [10]These authors contributed equally: Simone Scalabrin, Gabriele Magris, Mario Liva. ✉e-mail: digaspero@appliedgenomics.org; michele.morgante@uniud.it

PacBio reads in combination with high-coverage Illumina reads yielded a genome assembly with a contig N50 of 12.2 Kb and a scaffold N50 of 54.5 Kb, providing only a marginal improvement in contiguity as this assembly is still composed of 76,409 scaffolds[7]. The first attempt to use high coverage PacBio sequencing resulted in a significant improvement in sequence contiguity, yielding a genome assembly with a contig N50 of 1.3 Mbp and a scaffold N50 of 2.2 Mbp, but was still far from resolving the assembly of 2,684 scaffolds composed of 4,073 contigs into highly contiguous chromosome pseudomolecules[8]. More recently, a combination of PacBio, 10x Genomics and Illumina sequencing yielded the most complete assembly available so far, consisting of a contig N50 of 3.9 Mbp and a scaffold N50 of 42.5 Mbp, with 22 chromosomes resolved in 22 scaffolds, but with 2,810 additional unplaced scaffolds still containing one tenth of the assembled sequence (GenBank Assembly Accession GCA_003713225.1).

Accurate and complete reference genomes are necessary for disentangling complexities in polyploid genomes and obtaining reliable population genetic data from reference-guided resequencing analyses. Reduced-representation genome sequencing of *C. arabica* germplasm has revealed that the entire species has very low nucleotide diversity. Variant sites among *C. arabica* accessions are not shared with the progenitor species, as if both spontaneous and cultivated *C. arabica* carry only a set of recent mutations that started to accumulate after a recent speciation event[6,9]. Genetic and phenotypic differentiation emerged between two major groups represented by either Ethiopian or Yemeni accessions[10]. Ethiopian accessions have remained confined in Africa until recently with a few exceptions (e.g. the variety Geisha). A group of so-called Yemeni accessions, collectively known as Bourbon/Typica and considered to have originated in Ethiopia, were introduced in other coffee-producing countries during the past three centuries. Mutants within this narrow genetic base have long remained the major source of phenotypic variation for Arabica coffee production.

The narrow genetic basis of *C. arabica* was first broadened in the first half of the past century thanks to the spontaneous occurrence of an Arabica × Robusta hybridization in East Timor that generated a descent group of so-called Timor hybrid derivatives displaying rust resistance and heterosis–later exploited by intentional breeding in Portugal and Latin America[11]. Other potential opportunities for enlarging the cultivated Arabica gene pool include the exploitation of spontaneous hybrids between *C. arabica* and *Coffea liberica* in India[12,13], the newly formed Arabica × Robusta hybrids in New Caledonia that have remained confined locally[14,15] and hybridization breeding using *Coffea racemosa*[16].

Cytogenetic evidence suggests that irregularities in disomic meiotic behavior are ordinary in *C. arabica*, possibly due to pachytene chromosome secondary associations between homoeologs[17] and to anaphase chromosome lagging[18]. Univalent, trivalent and quadrivalent chromosome associations are reported to occur in gamete mother cells of the cultivars SL28 and Caturra with frequencies that are not negligible[18,19]. The frequency and the extent of DNA content variation that are observed among phenotypic mutants within cultivars could provide support for the presence of aneuploidies in *C. arabica* cultivated germplasm[20]. While aneuploidy in general has lethal or negative effects on plant development and growth in diploid species, in the case of a polyploid species it is more tolerated and in a species with very low diversity it may be even positively selected in the quest for phenotypic variation. This notion finds support in the fact that phenotypic variants that emerge among regenerated plants following somatic embryogenesis are almost systematically found to have altered chromosome numbers, while wild-type plants have a normal chromosome number[21,22].

In this work, we use ONT long reads and Hi-C data to generate a chromosome-scale assembly of *C. arabica* that represents a clear improvement over earlier versions for contiguity and completeness as well as for consistency in order and orientation with the assemblies of the diploid progenitors *C. canephora* and *C. eugenioides*. We utilize this resource to describe the structure of *C. arabica* chromosomes, including previously inaccessible pericentromeric regions, and identify reciprocal and nonreciprocal homoeologous exchanges compared to their parent-of-origin species that occurred early in the establishment of the polyploid lineage and are today shared by the entire species. We also refine, with the support of publicly available whole-genome resequencing data[23,24], earlier estimates of genetic diversity in the *C. arabica* species. Thanks to our analyses we are able to reconcile opposing pieces of evidence that claim the existence of substantial phenotypic diversity in *C. arabica* in the absence of high nucleotide diversity. We show that different types of chromosomal aberrations and exchanges between homoeologous chromosomes are frequent and happened not only right after the polyploidization event but, contrary to assumption, also in much later phases and that they remain polymorphic in the population. Finally, we see evidence of cryptic Robusta introgression via Timor hybrid derivatives in Arabica-like germplasm (i.e. accessions/entries that are taxonomically assigned to *C. arabica* in genetic or genomic repositories but show evidence of *Coffea* sp. introgression in genomic analyses).

## Results and discussion

### Contiguity, accuracy and completeness of the genome assembly

The assembled sequence of *C. arabica* amounts to 1.32 Gbp (Supplementary Table 1), nearly matching the expected genome size of 1.33 Gbp that was based on the observed 2 C value of 2.71 pg ± 0.04 in flow cytometry assays[5]. Of these, 1,098,789,244 bp of ungapped sequence were assembled into 22 chromosome pseudomolecules, consisting of 175 contigs, 80 scaffolds and 22 superscaffolds (Fig. 1a, Supplementary Table 2 and Supplementary Method 1). The assembly was compared to BAC sequences to assess the accuracy of the consensus sequence. We found an overall 99.4% sequence identity between the assembly and BAC scaffolds across a sample of 1.5 Mbp over 350 random genomic regions. This assembly shows more completeness and contiguity as well as better consistency in order and orientation with the assemblies of the diploid progenitors than previous *C. arabica* assemblies (Supplementary Figs. 1-11 and Supplementary Tables 3-4).

We predicted 57,794 gene models with a median length of 2,409 bp and a median number of 4 exons. Of these, 27,337 genes with a median length of 2,528 bp belonged to the canephora subgenome. Another 28,197 genes with a median length of 2,480 bp belonged to the eugenioides subgenome. As few as 2,260 genes were predicted in unanchored scaffolds. The cumulative length of coding sequences amounted to 85.5 Mbp, representing 6.5% of the total genome length. Introns showed a median length of 238 bp and covered cumulatively 12.5% of the genome length. BUSCO analysis[25] indicated that 99.3% of the expected universal single-copy genes were completely assembled and 0.7% were not present in the assembled sequence, neither as fragmented nor as partial copies. As expected in a tetraploid species with a relatively recent origin[6], 91.5% of the expected universal orthologs were complete and duplicated.

Telomeric repeats were assembled at the termini of the chromosome pseudomolecules in 38 out of 44 chromosomal ends (Fig. 1a). The assembly of the lower end of Chr7e and the upper end of Chr11e was interrupted by the distal presence in the pseudomolecule of partially assembled 35S rDNA arrays (35 and 30 repeats, respectively). The assembly of their homoeologous chromosomes (Chr7c and Chr11c) was interrupted proximally to the start of the 35S rDNA array. One unanchored scaffold (scaffold_682, 127 Kb in size) contained telomeric repeats at one terminus and twelve 35S rDNA units at the other terminus. The orientation of the telomeric repeats and the rDNA in scaffold_682 in relation to the orientation of the same sequences in chromosomes 7 and 11 makes us believe that it may represent the chromosomal end of either Chr7c or Chr7e, providing evidence for the

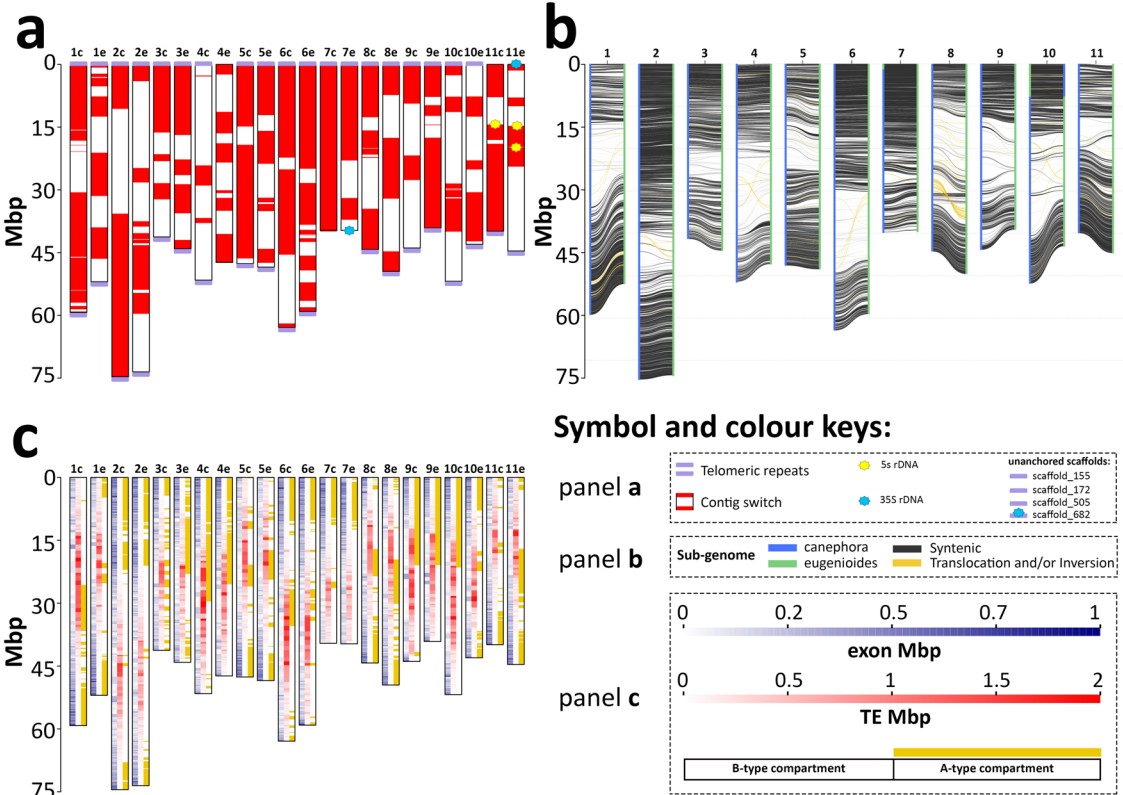

**Fig. 1 | Graphical representation of *C. arabica* chromosomes. a** Contiguity and completeness of the genome assembly and location of rDNA arrays. **b** Synteny plot between *C. arabica* subgenomes. **c** Gene and transposable elements (TE) density across 4,467 non-overlapping genomic windows corresponding to 100 Kb of non-repetitive DNA and A/B type chromatin compartments. In all panels, y-axis indicates million base pairs (Mbp). Source data are provided as a Source Data file.

subtelomeric location of the 35S rDNA locus on chromosome 7 in *C. arabica* (Fig. 1a). 290 additional repeats of the 35S rDNA unit were found in 16 unanchored scaffolds. The 35S rDNA therefore amounted to ~3.42 Mbp and consisted of a conserved ~9.5 Kb monomer. The 35S rDNA arrays were interrupted by 3 transposable element (TE) insertions. Two 5S rDNA loci were assembled on Chr11e and one on Chr11c (Fig. 1a). Chr11 is therefore carrying both 35S rDNA arrays and 5S rDNA arrays. Two additional 5S rDNA loci were expected on two other chromosomes based on literature reports of FISH assays[26] and were not assembled into any of the chromosome pseudomolecules. Additional 5S rDNA repeats were found in 28 unanchored scaffolds. 5S rDNA amounted cumulatively to ~4.58 Mbp and consisted of a conserved ~0.5 Kb monomer. Telomeric repeats were missing from the Chr4e pseudomolecule. Three unanchored scaffolds consisted almost entirely of telomeric repeats. A near-complete and contiguous sequence was obtained in one chromosome (Chr7c), resolving centromeric and pericentromeric tandemly repeated satellite arrays (Figs. 1a and 2a).

### Chromatin organization and evolutionary history of the *C. arabica* genome

The availability of a highly accurate and complete assembly of an allotetraploid genome allows for an accurate comparison of the two homoeologous genomes to achieve a better understanding of their evolutionary and functional dynamics. The same Hi-C data obtained from young leaf tissues that was used for the scaffolding of the assembly was also used, after mapping of the reads onto the chromosome pseudomolecules, to study the three-dimensional chromatin organization. Similarly to what has been observed in the larger barley genome[27], we observed the presence of a recognizable anti-diagonal pattern in the intrachromosome Hi-C contact matrices (Supplementary Fig. 12), indicating the occurrence of a RABL configuration in

*C. arabica* interphase nuclei where the two chromosome arms are in contact with one another following their folding at the centromere. A higher contact frequency among subtelomeric regions, as observed in barley[27] and Arabidopsis[28], is also evident from the Hi-C contact matrix.

We performed a principal component analysis on distance-normalized interaction matrices at 50 Kb resolution and used the sign of the PC1 values to assign 100 Kb genome windows to either the active and less compact A compartment (usually corresponding to loose and highly-transcribed euchromatic zones) or to the inactive and more compact B compartment (usually corresponding to tightly-packed heterochromatic zones)[29]. In total, we observed that the *C. arabica* genome in the nuclei of young leaves comprises approximately 465 Mbp (44.2%) that correspond to the A compartment and 586 Mbp (55.8%) that correspond to the B compartment. These relative proportions corresponding to A and B compartments are very similar in the two subgenomes (A representing 44.1% of the canephora and 44.3% of the eugenioides subgenome, respectively) and the large scale chromatin organization is frequently very similar between the two homoeologous chromosomes (Supplementary Figs. 13-16). A limited number of chromosomes present A compartments at both ends flanking a large B compartment while the vast majority of chromosomes presents an asymmetric chromatin organization with the A compartment occupying only one end of the chromosome (Supplementary Figs. 13-17). A and B chromatin compartments show different structural and functional attributes as well as a very different evolutionary history.

Structurally, chromatin compartments track repeat and gene density quite closely (Fig. 1c and Supplementary Fig. 17). A compartments are repeat poor and gene rich, while B compartments show the opposite organization (Supplementary Fig. 18a, b). All superfamilies of transposable elements are located predominantly in B compartments

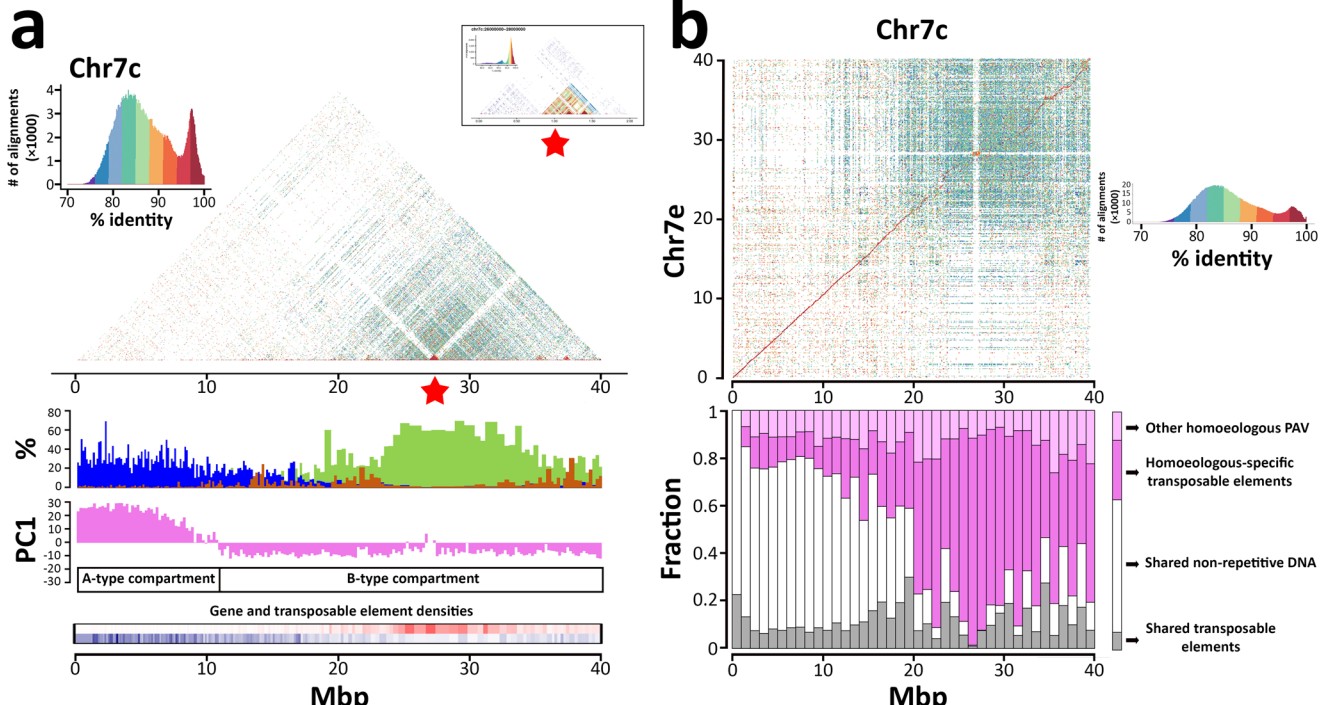

**Fig. 2 | Sequence and structural variation between homoeologous chromosomes in euchromatic and heterochromatic regions. a** Composition and chromatin organization of Chr7c. Sequence identity heatmap on top of the panel with a tandem repeat array of a highly conserved 2.7 Kb monomer indicated by the red asterisk and magnified in the inset; histograms of gene (blue), Athila- (brown) and chromovirus-derived (olive green) sequence abundances, expressed as percentage of exonic base pairs (genes) and percentages of masked base pairs by RepeatMasker using intact *Coffea* Athila and Chromovirus sequences across 4,467 non-overlapping genomic windows containing 100 Kb of non-repetitive DNA and histograms of PC1 values defining A/B compartments using non-overlapping genomic windows of 100 Kb. **b** Comparison between Chr7c and Chr7e. Each dot represents sequence alignments with >70% of identity between non-overlapping 2 Kb windows. The color of each dot represents the % of sequence identity. Box plots represent the fraction of nucleotides shared between (white and gray) or private to (pink and magenta) the homoeologs. These categories are further sorted into the fraction of nucleotides in annotated transposable elements (gray and magenta) and in non-repetitive DNA (white). The pink stack includes low-copy DNA as well as other DNA tracts that are not annotated as transposable elements outside of collinear regions. In both panels, x-axis indicates million base pairs (Mbp). Source data are provided as a Source Data file.

with differences among superfamilies in the relative abundance between A/B compartments (Supplementary Table 5).

Evolutionarily, A compartments have significantly more shared sequence and higher sequence identity between subgenomes than B compartments (Fig. 2b and Supplementary Fig. 18c-f), mainly as a consequence of a different history of TE insertions. As a whole, the two subgenomes within *C. arabica* are collinear across 444 Mbp including 132 Mbp of shared TEs, cumulatively accounting for 40.8% of the canephora subgenome and 40.1% of the eugenioides subgenome, with an average sequence identity of 94.9% (Fig. 1b, c and Supplementary Figs. 19-24). The remaining parts of the subgenomes are highly enriched in structural variants due to presence/absence variation of TEs that is part of the standing variation between the two diploid progenitor species, involving cumulatively 519 Mbp and corresponding to 47.3% of the total length of the chromosomes (Supplementary Figs. 1-11 and 19-24). 135 Mbp of non-shared sequences due to presence/absence variation that could not be attributed to known TEs, corresponding to 12.3% of the total length of the chromosomes, are also present (Supplementary Table 6). We see a marked difference in the proportion of shared sequence when comparing A and B compartments, which is not due to low/single copy DNA sequences, but rather to the large number of non-shared TE elements that are present in the B compartments. Dating of the LTR-retrotransposon insertions on the basis of the intraelement LTR divergence shows that the shared retroelements are significantly older than the non-shared ones (Supplementary Fig. 25 and Supplementary Method 2). This is consistent with the hypothesis that the shared ones predate the separation between the two ancestral species and the non-shared ones represent younger elements that

have inserted after the separation, contributing to a dramatic sequence divergence that is visible across large chromosomal regions corresponding to B chromatin domains (Supplementary Figs. 1-11 and 19-24). The very recent origin of *C. arabica* does not allow us to determine if at least some of the non-shared insertions occurred after the allopolyplodization event and could contribute to the standing sequence variation in *C. arabica*.

Functionally, A compartments are transcriptionally more active in all organs analyzed (Supplementary Fig. 26). A previous analysis of expression differences between homoeologous genes shows similar levels of expression for 65% of gene pairs[30]. We related differences in expression level between homoeologous genes to the chromatin compartments. All tissues analyzed showed greater differences in expression levels for gene pairs located in B compartments than in A compartments (the differences were statistically significant in 10 of the 12 samples analysed) that could result from the higher levels of structural variation present in B compartments that could lead to variation in cis-regulatory elements acting on the homoeologous copies of the genes (Supplementary Fig. 27).

We performed a detailed comparative analysis of structural, functional and evolutionary aspects between the two ancestral genomes for a group of genes involved in the caffeine biosynthetic pathway that are important for *Coffea* cultivation and coffee production. The loci containing specific N-methyltransferases that are involved in the control of three key steps in the caffeine biosynthetic pathway[31] are located on *C. arabica* chromosomes 1 and 9. The 3,7-dimethylxanthine N-methyltransferase (*DXMT*) gene, which controls the final step of enzymatic conversion of theobromine into caffeine and whose

deletion or reduced expression explained variation in caffeine synthesis in *Coffea humblotiana*[32] and in *C. arabica*[33,34], is located in the upper subtelomeric region of Chr1. The canephora homoeolog in *C. arabica* (Chr1c) carries a single *DXMT* copy (Supplementary Fig. 28a). The eugenioides homoeolog in *C. arabica* (Chr1e) carries a ~6.5 Kb-long tandem duplication including the gene body and part of its 5' intergenic region that generated two *DXMT* copies (Supplementary Fig. 28b). The canephora-derived *DXMT* is expressed in *C. arabica* in multiple organs (i.e. leaf, stem, root, bud, shoot apical meristem and drupe). The expression of the canephora-derived *DXMT* showed the highest level in developing green drupes and the lowest in fully ripe red drupes and in bulked drupe samples collected at intermediate ripening stages (Supplementary Figs. 28 and 29). The eugenioides-derived *DXMT* paralogs are expressed substantially only in developing drupes during their green stage (Supplementary Figs. 28 and 29). Despite *DXMT*s being located within a gene rich and repeat poor subtelomeric A chromatin domain, the intergenic space around *DXMT* genes is characterized by the presence of homoeologous-specific repetitive DNA that may account for the observed differences in organ-specific regulation of gene expression (Supplementary Figs. 30 and 31). Xanthosine methyltransferase (*XMT*) and 7-methylxanthine N-methyltransferase (*MXMT*)—two genes that control the synthesis of theobromine and its precursors—colocalize in a gene cluster containing several xanthine-methyltransferases (*XMT*) on both Chr9c and Chr9e. The cluster on Chr9c contains 5 *XMT* predicted genes in a 153.1 Kb region (Supplementary Fig. 32a). The cluster on Chr9e contains 6 *XMT* predicted genes and translated nucleotide sequences with similarity with *C. canephora MXMT* and *XMT* proteins[31] in a 191.5 Kb region (Supplementary Fig. 32b). The region spanning the *XMT* cluster shows little collinearity outside of coding sequences between homoeologs in *C. arabica* and with *C. humblotiana* (Supplementary Fig. 33). Within this gene cluster, the paralog controlling the enzymatic conversion into theobromine (*MXMT*, corresponding to the *Cc00_g24720* gene in *C. canephora*) is expressed in *C. arabica* organs at similar levels from the two subgenomes. As for the other *XMT* gene copies in *C. arabica* homoeologous clusters, the ones that show the highest sequence similarity with the *XMT* copy that is expressed in *C. canephora* seeds (*Cc09_g06970*) are also the ones that are more expressed in *C. arabica* organs (Supplementary Fig. 29). The canephora-derived homoeolog of this *XMT* copy is transcribed at much higher levels than the eugenioides-derived *XMT* homoeolog in all organs and replicates, a situation that is most extreme in developing green drupes. Expression patterns in drupes for *XMT* and *MXMT* are largely in agreement with a recent detailed analysis of expression of these genes in coffee beans[35] while for the *DXMT* genes the differences between the genes in the two subgenomes appear to be less dramatic when the cumulative expression of both copies present in eugenioides is considered. As previously highlighted[35], the expression patterns of these caffeine biosynthetic genes appear to be highly correlated with caffeine content in mature seeds.

## Centromeric regions of the *C. arabica* genome

In contrast to previous statements on the absence of centromeric satellite arrays in *C. arabica*[36], whose presence is otherwise common in plant centromeres, we detected large satellite structures of tandem repeats in *C. arabica* that reached individually several hundred Kb in size and collectively extended for several Mbp across the B chromatin compartments in each chromosome (Supplementary Figs. 17, 34-35). We did not detect short monomeric units (100-200 bp) forming higher order repeated structures that are usually observed across kingdoms (i.e. in budding yeast[37], *Arabidopsis*[2], and humans[38]) and promote centromere formation. No tandemly arranged regions with repeat units shorter than 2 Kb in size were found using Tandem Repeats Finder[39]. A considerable portion of the heterochromatic regions of each chromosome consisted, instead, of large arrays of tandem

repeats with units longer than 2 Kb, whose sequences are usually derived from different chromodomain-containing Gypsy retrotransposons (Fig. 2a and Supplementary Fig. 17). CRM, Tekay, Galadriel, Reina and Athila derived-sequences account for 45.4% of the *C. arabica* genome and are specifically localized in the B chromatin compartment of each chromosome (Supplementary Fig. 17). Monomeric units mostly derived from the CRM and Tekay clades and, to a more limited extent, from the Reina and Galadriel clades form complex arrangements of alternating arrays along the chromosome (Supplementary Figs. 34-35). The large chromosomal regions containing the highest density of chromovirus-derived arrays are usually flanked by regions with the highest densities of Athila-derived tandemly arranged arrays that are sometimes interspersed with chromovirus-derived arrays (Supplementary Fig. 35). In Chr7c, which consists of a single contig that should span the whole centromeric region (Fig. 1a), chromovirus-derived arrays and Athila-derived arrays are interrupted by a 690 Kb-long satellite structure formed by a 2,683 bp CRM-derived monomer that is highly conserved in sequence composition within the array and highly diverse in sequence composition from the surrounding region (Fig. 2 and Supplementary Method 3). The same monomer forms a 394 Kb-long array in the same region of the homoeologous chromosome (Chr7e). The phylogenetic analysis of the monomeric units indicates that the formation of these arrays likely predates the divergence between the two progenitor species. Their patterns of intra-array sequence variation also suggest that they expanded and evolved following different dynamics (Supplementary Fig. 36). The array on Chr7e shows large portions that are formed by highly similar monomers, as if parts of this array expanded more recently or underwent sequence homogenization through cycles of unequal crossover. The array on Chr7c shows lower sequence identity among monomers, with monomers that become gradually less similar in sequence as their physical distance increases. Both the Chr7e and Chr7c arrays are interrupted by insertions of LTR-retrotransposons (one in Chr7e, which was then duplicated, and 8 in Chr7c, Supplementary Figs. 37-38). The retrotransposon insertions occurred over a large time span ranging from 9.2 to 0.3 MYA (Supplementary Table 7). The older insertions are located in the distal part of the Chr7c array that also shows higher sequence divergence among monomers. No other chromosome pseudomolecule carries arrays of this monomer, but the same monomer is present in 27 unanchored scaffolds that amounted collectively to 4.0 Mbp and contained tandem arrays of this monomer collectively covering 97.9% of their sequence. These scaffolds could represent centromeric satellite arrays belonging to other chromosomes that were not assigned to chromosome pseudomolecules. The complex architecture of the *C. arabica* B compartments and the lack of tandem repeats of a single short monomer across all chromosomes, such as the conserved human α-satellite and the *Arabidopsis CEN180* satellite[2] that promote the formation of the centromere in those genomes, preclude the prediction of the exact position of *C. arabica* centromeres using only DNA sequence features.

## Chromosomal rearrangements between subgenomes in the sequenced Bourbon genotype

Although the ancestral chromosomal location of homoeologous gene pairs has generally been maintained following the polyploidization event, the *C. arabica* assembly shows a few exceptions to this general rule. We precisely mapped the location of three subtelomeric events of exchange between homoeologous chromosomes[40], all involving A chromatin compartments and resulting in autotetraploid chromosomal regions (Chr2c:74,564,297-74,718,367, Chr7c:1-1,252,944 and Chr10e:1-56,401). The largest of these is illustrated in Fig. 1b, while the others are too small in size to be visible at the graphical magnification of Fig. 1b. These events have led to the replacement of canephora homoeologous chromosome segments with eugenioides homoeologous segments which collectively impacted 169 predicted genes

for which the allopoliploidy condition has been lost. These homoeologous replacements are fixed in *C. arabica* as shown by the lack of homoeologous SNPs in these regions in all accessions analysed (see next paragraph). The analysis of SNP distribution in the long reads supporting the assembly reveals that the homoeologous exchange on Chr2 occurred in the intergenic space between the genes *Cara0002c46780* and *Cara0002c46790* on Chr2c and *Cara0002e47690* and *Cara0002e47700* on Chr2e. The homoeologous exchange on Chr7 occurred within a Helitron transposable element while the homoeologous exchange on Chr10 occurred within a Mutator transposable element. The chromosome-scale assembly of 'Bourbon' has also revealed a previously undetected large event of homoeologous recombination involving a reciprocal and symmetrical exchange, possibly originating from a mitotic crossing-over[41] between parental chromosomes that occurred in an interspecific hybrid diploid cell lineage before the polyploidization event. The reciprocal exchange swapped 7.6-Mbp of canephora and eugenioides arms of the native chromosomes 10 (Fig. 1b). The site of reciprocal exchange occurred within the transcriptional unit of a kinesin-like protein KIN-14R (gene models *Cara0010e09280* and *Cara0010c09420*, shown in Supplementary Fig. 39). *Cara0010e09280* and *Cara0010c09420* are transcribed from the negative strand into 19 exons. An apparent crossing-over associated conversion tract (COCT)[42] of approximately 3 Kb is visible between exon 6 and the middle of intron 10 where both homoeologs present a nearly identical sequence tract that is more similar to the present-day *C. eugenioides* than to the present-day *C. canephora*. This reciprocal event of intragenic exchange neither disrupted the intron-exon structure nor affected the transcriptional activity of the gene (Supplementary Table 8). Using a PCR-based assay, we experimentally validated the evidence of this homoeologous recombination in 'Bourbon' and showed that this event is not restricted to the Bourbon/Typica germplasm of Yemeni origin, but is also present in the accession '1-Geisha' that is a representative of the Ethiopian germplasm, suggesting it is the result of an ancient event that predates the split between the Ethiopian and Bourbon/Typica germplasm (Supplementary Fig. 40). After this homoeologous exchange, a small subtelomeric homoeologous replacement event (described above) replaced 55 Kb of canephora DNA (previously moved on the Chr10e homoeolog) with eugenioides DNA.

Unlike the cases reported so far, the large intrachromosomal inversion and translocation that differentiates Chr8c and Chr8e (Fig. 1b, Chr8c:26,514,569-30,375,036, Chr8e:34,155,493-37,608,685) does not seem to be the result of a post-polyploidization event (see also supporting evidence of the contact map in Supplementary Fig. 12 and the collinearity with the same chromosomes in the Caturra assembly in Supplementary Figs. 1-11 to exclude assembly-related issues). Although the assemblies of the *C. canephora* and *C. eugenioides* genomes are highly fragmented over this chromosomal region (Supplementary Figs. 1–11), they seem to confirm that this intrachromosomal rearrangement is part of the standing variation between the two diploid progenitor species and thus that the chromosomal rearrangement predates the polyploidization event.

While the four previously described post-polyploidization chromosomal rearrangements appear to have occurred very early on during the *C. arabica* evolution and are present today in all accessions we surveyed, we identified an additional event of subtelomeric homoeologous replacement that is present in the Bourbon specimen from which we extracted genomic DNA for ONT sequencing but not in the Bourbon specimen from which we had previously extracted genomic DNA for Illumina sequencing[6]. The eugenioides to canephora replacement on the lower end of Chr1c, involving 5.7 Mbp (Supplementary Fig. 41) and occurring in A chromatin compartments like the other four we described earlier, has generated a heterozygous variant in the accession used for sequencing, which showed a consistently lower-than-average read coverage on Chr1c and a consistently higher-than-

average read coverage on Chr1e that are compatible with a condition of CAN:EUG = 1:3 copy number variation over that region. The pattern of homoeologous SNPs in continuous and split long-reads that span the exchange site indicates that the event occurred within the homoeologous genes *Cara0001c18950* on Chr1c and *Cara0001e19460* on Chr1e or immediately upstream. In the specimen previously used for Illumina sequencing we did not observe distortions from average read coverage on either homoeolog, suggesting a native CAN:EUG = 2:2 copy number condition. Hi-C data using DNA from the same specimen used for ONT sequencing showed a mixture of Chr1c/Chr1c, Chr1e/Chr1e and Chr1c/Chr1e contacts around the site of this homoeologous replacement (Supplementary Fig. 42), lending support to its existence and to its heterozygous state. The observation of this apparent de novo chromosomal replacement event that involves a large genomic region that is rich in genes suggests that chromosomal rearrangements between homoeologs of this type could represent an important mechanism for the generation of intraspecific genetic variation in a very recently originated species, where nucleotide diversity is still very low.

## Genetic diversity in the species *Coffea arabica* and in Arabica cultivated germplasm

We called 7,694,774 variant sites in a set of 174 *Coffea* sp. accessions out of an average of 716,303,048 informative sites per accession across the 22 chromosome pseudomolecules. Based on the taxonomic classification of publicly available accessions in their original literature reports[6,9,24] and its revision based on the analyses of this paper (Fig. 3a-d and Supplementary Fig. 43), our WGS panel consisted of 34 accessions of *C. canephora*, one accession of *C. eugenioides*, 95 accessions of *C. arabica* and 44 *C. arabica* × *Coffea* sp. introgression lines (Supplementary Data 1). *C. arabica* accessions had a median nucleotide diversity of $\pi = 3.12 \times 10^{-4}$ (Fig. 3c). Genomic windows with outlier $\pi$ values were scattered randomly across the genome in *C. arabica* (Fig. 3d). Cumulatively, *C. arabica* accessions showed very low numbers of homozygous SNPs per genomic window (Supplementary Fig. 44), in particular across the canephora subgenome which was more easily purged of false positive SNPs than the eugenioides subgenome.

Ethiopian accessions are confirmed to be differentiated from Bourbon/Typica accessions (Fig. 3b) and contribute a limited number of genomic regions with higher homozygous SNP frequencies. The distribution of heterozygous SNP counts in *C. arabica* is likely to reflect the baseline false positive error rate in SNP calling as most of these accessions are expected to be homozygous based on their mating system. The error rate could not be lowered with the filtering procedure because of the low coverage of publicly available sequencing data (Supplementary Data 1). We did not detect an increase in homoeologous SNP frequency in any of the accessions analysed in any of the three regions where we detected homozygous homoeologous replacement events in 'Bourbon', as would have been expected if some accessions contained both homoeologs for those regions. This provides evidence that the homoeologous replacements that we observed in the genome assembly of 'Bourbon' represent ancestral events. Progenitor diploid species showed values of nucleotide diversity that are one order of magnitude higher than *C. arabica* with a rather uniform genomic distribution (Fig. 3c,d), with the exception of *C. eugenioides* that is represented here by a single accession with tracts of homozygosity. We detected 1,877,440 SNPs in 95 bona fide accessions of *C. arabica* that did not show signatures of interspecific introgression (Supplementary Data 1). The phylogenetic tree obtained with this dataset showed strongly supported branches (bootstrap values > 80%) that predominantly include either widely used cultivars of the Bourbon/Typica group, Ethiopian germplasm locally exploited by the forest coffee production system, or Ethiopian germplasm locally exploited by the garden coffee production system (Supplementary Fig. 45).

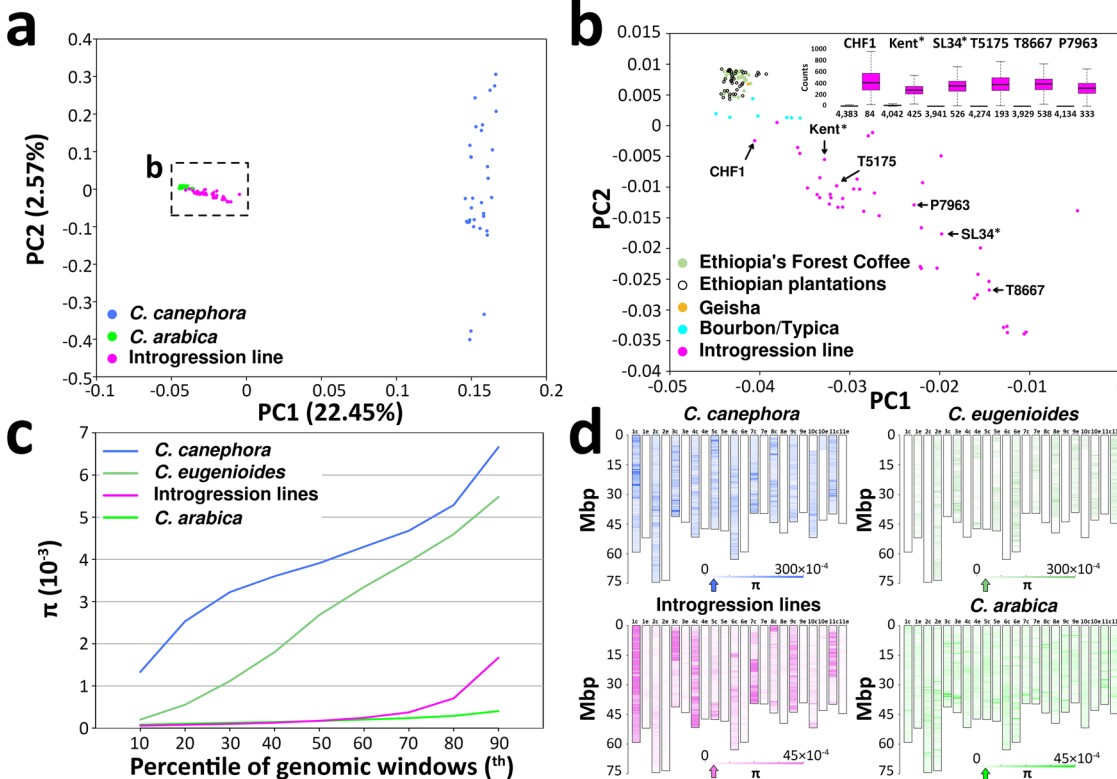

**Fig. 3 | Genetic diversity in 172 *Coffea* sp. accessions. a** Bidimensional plot of the first two components of a Principal Component Analysis (PCA). It has to be noted that the accession of *C. eugenioides* is not included in **a** and it is provided in Supplementary Fig. 43 and the accession *C. canephora* 33-1 has been removed from both PCAs due to low coverage (Supplementary Data 1). **b** Magnified view of the section of the bidimensional plot delimited by the rectangle in **a**. **b** (inset) Box plot distribution of the number of SNPs per genomic window of 100 Kb of non-repetitive DNA sorted based on residing in regions free of *C. canephora* introgression (olive green boxes to the left of each variety) or in regions carrying *C. canephora* introgression (magenta boxes to the right of each variety) in 3 Arabica-like specimens as well as 3 Timor hydrid derivatives (P7963, T5175 and T8667).

Boxes indicate the first and third quartiles, the horizontal line within the boxes indicates the median and the whiskers indicate ±1.5 × interquartile range. Numbers below each box indicate the number of genomic windows in each genome compartment. Percentile distribution (**c**) and chromosomal distribution (**d**) of nucleotide diversity (π) across 4,467 non-overlapping genomic windows containing 100 Kb of non-repetitive DNA. In **d**, the arrows on the bar scales point to median values, y-axes indicate million base pairs (Mbp). *The sequenced specimens BioSamples SAMN10411969 and SAMN10411970 are indicated here as Kent and SL34, respectively, consistent with their SRA metadata. Source data are provided as a Source Data file.

The tree also showed a clear geographic patterning of the Ethiopian genetic diversity. Within this geographic pattern, cultivars of the Bourbon/Typica group, also referred to as Yemeni accessions, are grouped in a statistically supported branch with landrace varieties from the Ethiopian highlands east of the Main Rift (Supplementary Fig. 45). All the accessions that we have reclassified as *C. arabica × C. canephora* introgression lines (Supplementary Data 1) present a predominant fraction of their genomic windows that show nucleotide diversity estimates comparable to *C. arabica* and a variable fraction of their genomic windows (depending on the accession) in the canephora subgenome that show estimates comparable to those observed within *C. canephora*. The genomic windows in *C. arabica × C. canephora* introgression lines that show elevated nucleotide diversity are not randomly distributed across the genome but are found in large continuous chromosome segments that are compatible with the presence of *C. canephora* introgressed haplotypes. As a result of this, *C. arabica × C. canephora* introgression lines appear to be displaced on the PCA biplot from pure *C. arabica* germplasm represented by Ethiopian accessions and Bourbon/Typica varieties, which all together encompass very low diversity (Fig. 3b). Individual *C. arabica × C. canephora* introgression lines that lie increasingly apart from the core of the pure *C. arabica* germplasm differ in the number of putatively introgressed genomic windows but not in the levels of nucleotide diversity in these genomic windows. All of this is true for both known *C. arabica × C. canephora* introgression lines (Supplementary Data 1),

represented in Fig. 3b by the Timor-hybrid derived (Catimor) accessions P7963, T5175, T8667, as well as for a group of 41 other sequenced specimens previously classified as Arabica in their literature reports and in their archived metadata[23,24]. In the original literature reports, some of these accessions were assigned codes and countries of origin suggesting that they were either deliberately sampled as introgression lines or were introduced from germplasm repositories of breeding institutions (Supplementary Data 1). The introgressed haplotypes in those accessions (*n* = 37) tend to extend over shared chromosomal regions (Supplementary Fig. 46 and Supplementary Method 4) as if they were preferentially retained due to selective advantages (Supplementary Data 2) and/or they derived from a common ancestral event (Supplementary Figs. 47 and 48). An exception is represented by the accession S288 that, based on the code it was given, was expected to correspond to an unrelated *C. liberica* introgression line (Supplementary Fig. 49). African and Indian specimens carrying introgressed genomic windows also include 3 renowned Arabica-like varieties (i.e. Kent, SL28 and SL34), 2 accessions sampled from Ethiopian plantations (GNG1 and GUG3) and one accession from the Ethiopia's forest production system (CHF1)[23]. Kent, SL28 and SL34 are generally considered part of the native African crop germplasm. The accession CHF1 is the most peculiar case because it carries the lowest number of genomic windows with signatures of *C. canephora* introgression and this introgression is homozygous (Fig. 3b). These putative *C. arabica × C. canephora* introgression lines show a similar genome fraction with

evidence of *C. canephora* introgression as that found in Timor hybrid derivatives and they also share the location of the introgressed regions (Supplementary Table 9) as well several recombination breakpoints (Supplementary Fig. 50). These similarities could indicate that they have all inherited the *C. canephora* introgression from a common ancestor. The publicly available sequencing data for Kent and SL34 shows that the single specimen that was sequenced as a representative of each variety carries tracts of the *C. canephora* introgression in a heterozygous state, indicating either that their introgressions are not yet fixed in the seed pool representative of these varieties or that they may derive from local gene flow in the site from which the sequenced accessions were sampled. The publicly available sequencing data for SL28 shows that the single specimen that was sequenced as a representative of this variety carries tracts of the *C. canephora* introgression in a homozygous state (Supplementary Fig. 51). We resequenced a different accession of SL28 obtained from the CATIE coffee germplasm collection in Costa Rica (Supplementary Table 10), which showed no signatures of introgression (Supplementary Fig. 51), confirming that the introgression events that are detectable in individual accessions are likely due to recent contamination events of their present-day seed pool, posing threats to the genetic purity if new plantations are established using sexually propagated material from these sources. In order to understand the scale of this issue for the coffee industry and for the coffee scientific community and to verify that its emergence occurred after the worldwide spread of Timor hybrid and *C. liberica* derivatives, we reanalyzed GBS-data of ecotypes collected by prospections that were conducted in Ethiopia in the 1960's before the global spread of leaf rust resistant material and before its introduction in Ethiopian coffee production systems[6] (Supplementary Method 5). We compared these ecotypes with other accessions of cultivated varieties and landraces sampled in Ethiopia and elsewhere in and outside Africa, mostly held at the same gene bank of CATIE[6]. As expected, we did not find signatures of introgression in Ethiopian ecotypes predating the hypothetical date of arrival of leaf rust resistant material[43] (Supplementary Figs. 52-53, Supplementary Data 3 and Supplementary Method 6). On the contrary, we found cryptic Timor hybrid introgression in one accession of the Bourbon/Typica variety 'Laurina' that was introduced from Camerun but not in another accession of the same variety 'Laurina' that was introduced from Bourbon Island (Supplementary Fig. 54). We also found introgression from *C. liberica* in two *C. arabica* landraces from Tanzania (Supplementary Figs. 52-53, 55). In consideration of the fact that the *C. canephora* introgressions present in the Catimor lines are derived from the Timor hybrid, it is tempting to conclude that this event of spontaneous hybridization is contributing to the standing variation within Arabica germplasm, beyond its intentional use in breeding programs, and the same is occurring for other recent *C. arabica* × *Coffea* sp. hybridization events. After removing all these accessions with both known and cryptic introgression from WGS and GBS datasets (Supplementary Fig. 53), the analysis of genetic diversity of the curated set of *bona fide C. arabica* confirmed that cultivars, landraces and spontaneous germplasm all have a limited genetic base.

### Chromosomal aberrations and homoeologous exchanges as drivers of genetic diversity in the *C. arabica* species

Based on the identification of a putative recent homoeologous exchange event in heterozygous condition in the Bourbon individual that was used for ONT sequencing (Supplementary Fig. 56), we set out to identify additional events of both balanced as well as unbalanced structural variation (Supplementary Method 7) in the set of 94 *bona fide C. arabica* accessions that were used also for SNP analysis (Supplementary Data 1) and in 4 control accessions (Supplementary Table 10).

The comparison of read depth of coverage and homoeologous SNP variant frequency between each of 98 resequenced accessions and 'Bourbon'[43] (exemplified in Fig. 4) allowed us to identify aberrations of different types such as aneuploidies (Fig. 4a and Supplementary Fig. 57), deletions and duplications (Fig. 4b, Supplementary Fig. 58 and Supplementary Data 4) and homoeologous exchanges (Fig. 4a,b, Supplementary Figs. 59 and 60 and Supplementary Data 5) that were not shared with 'Bourbon'. Given the relatively low read coverage of most accessions, we could only reliably detect events that were at least 200 Kb in size and we could not perform an in-depth analysis of the individual chromosomal sites where the rearrangements took place. The approaches we used did not allow us to detect additional reciprocal exchanges such as the one identified in 'Bourbon' in chromosome 10 or other balanced chromosomal rearrangements because only events that changed the relative copy number between the two subgenomes could be identified. The main approach we used was based on changes in homoeologous variant frequency from the expected 50% value when two copies of each subgenome are present. We previously termed this approach Reduction Of Heterozygosity (ROH)[43] and we renamed it here Reduction of Homoeologous Heterozygosity (ROH$_H$). By combining the ROH$_H$ results with the analysis of relative Depth Of Coverage (DOC) in comparison to 'Bourbon' we were able to identify chromosomal aberrations and homoeologous exchanges both when they were in homozygous or heterozygous condition, as well as when they were apparently showing somatic mosaicism. This analysis allowed us to confirm previous observations[40] that the exchange events present in 'Bourbon' in chromosomes 2, 7 and 10 occurred very early on in the *C. arabica* lineage because they appear to be fixed in all the accessions we analysed (this was also confirmed by our SNP analysis, see above). However, we also detected 4 accessions showing whole chromosome aneuplodies, 44 accessions showing additional homoeologous exchanges to those already observed in 'Bourbon' and 3 accessions showing additional large deletions or duplications to those that may be present in 'Bourbon' (Supplementary Figs. 57 and 60). The vast majority of the events we detected fit the expectations for germinal mutations in heterozygous or homozygous state, both in terms of ROH$_H$ as well as in terms of DOC. However, there were two events, one involving aneuploidies and the other involving a homoeologous exchange that, while statistically highly significant and physically very large, could only be explained either by assuming that the individual sequenced underwent a somatic mutation that led to somatic mosaicism for the chromosomal aberration or that the sequenced material derived from at least two genetically heterogeneous individuals differing for the detected aberration.

We identified 3 accessions with trisomies, two for chromosome 9—one involving an additional copy of the eugenioides-derived chromosome and the other one an extra copy of the canephora-derived chromosome—and one for chromosome 11 involving an an additional copy of the canephora-derived chromosome. We also identified a putative chimerism in the accession MESF1 that involves trisomy for Chr2e and monosomy for Chr10c. Plants have long been known to be generally more tolerant of aneuploidy than animals[44] and polyploids do frequently present spontaneous occurrences of aneuploid individuals[45], presumably as a consequence of errors in properly partitioning the multiple chromosome sets during meiosis. The allohexaploid *Triticum aestivum* is known to tolerate monosomic and trisomic variants and newly created synthetic hexaploid wheat plants present an even higher frequency of aneuploidy[46]. The identification of aneuploid *C. arabica* accessions is therefore not unexpected. The aneuploid condition is not stably inherited through meiosis but vegetative propagation may favor long term maintenance of this condition. Karyotyping of the accessions involved may provide definitive proof of the detected aneuploidies. The presence of aneuploidies is in agreement with the observation of irregularities in disomic meiotic behavior in *C. arabica*[13,14] as well as with the observation of DNA content variation among phenotypic mutants within cultivars[22].

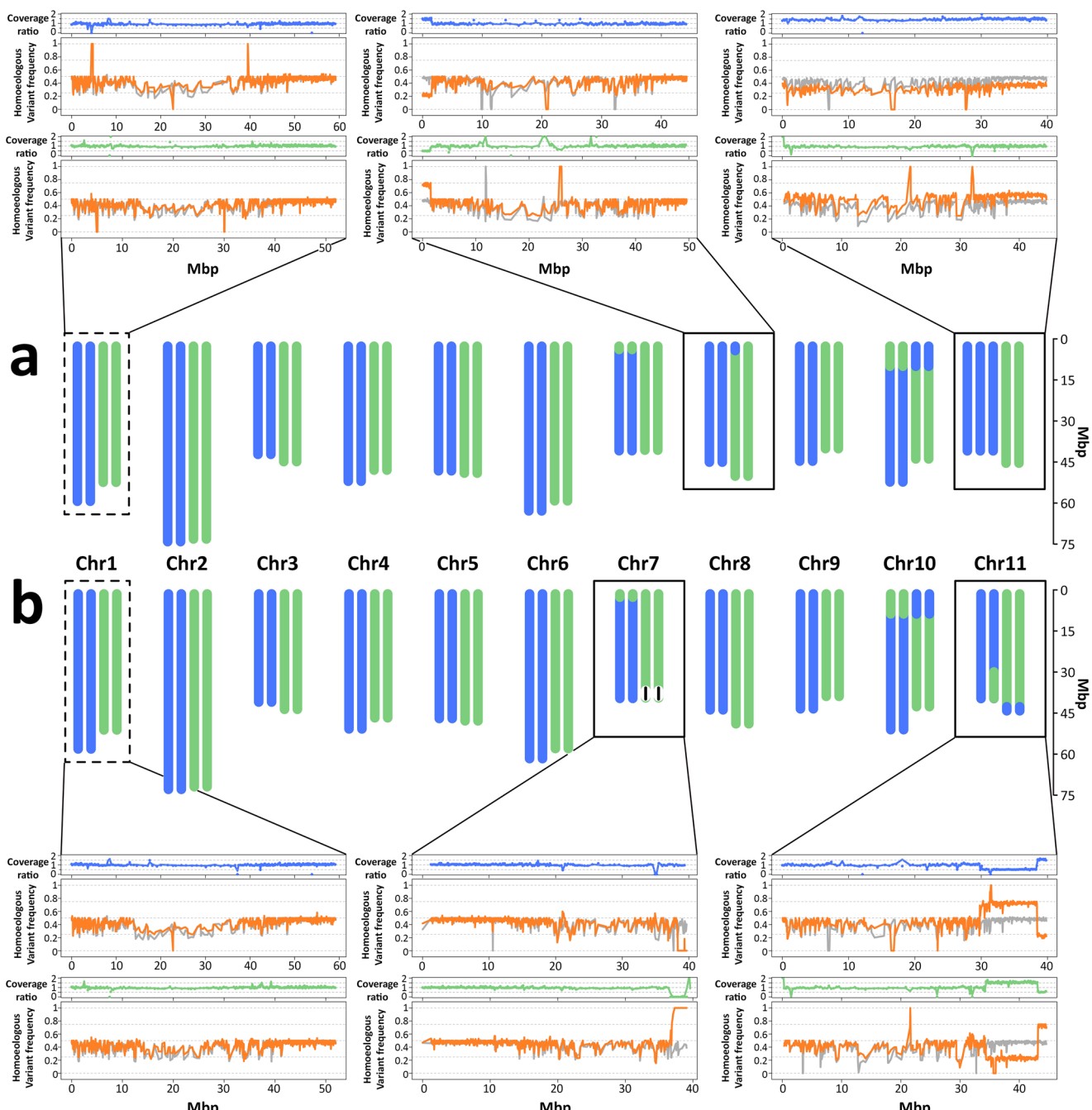

**Fig. 4 | Graphical representation of *C. arabica* karyotypes. a** GNG5 accession. **b** GISF2 accession. Blue and green vertical ideograms represent canephora and eugenioides homoeologous copies, respectively. Chromosomes, deletions and exchanges are drawn to scale. Plots next to each karyotype illustrate inter- and intra-chromosomal variation in depth of coverage of each homoeolog (canephora, blue line; eugenioides, green line) with respect to 'Bourbon' as well as homo-eologous variant frequency in either accession (orange line, forefront) and in 'Bourbon' (gray line, background) referenced to the canephora subgenome (above in each panel) and the eugenioides subgenome (below in each panel), in the chromosomes affected by trisomy, homoeologous exchange and large deletions

(highlighted by black line boxes) as well as in an unaffected chromosome (Chr1, dashed line boxes). Horizontal dashed gray lines forming the grid of the depth of coverage plot represent expected values under the conditions of 4:0 (y = 2), 3:2 (y = 1.5), 2:2 (y = 1) and 0:2 (y = 0) homoeologous copies. Horizontal dashed gray lines forming the grid of the homoeologous variant frequency plots represent expected values under the conditions of 4:0 (y = 1), 3:1 (y = 0.75), 2:2 (y = 0.5), 1:3 (y = 0.25) and 0:2 (y = 0) homoeologous copies. X-axes indicate million base pairs (Mbp). The complete series of chromosome plots in a graphically expanded version is available in the figshare repository[43].

Homoeologous exchanges were identified in a large number of accessions and all involved terminal portions of chromosomes 1, 5, 6, 7, 8, 10 and 11 (Fig. 5 and Supplementary Data 5) that were enriched in exonic sequences, depleted in TEs and more collinear between homoeologs than the average of the chromosome in which they have occurred (Supplementary Data 6). Collectively, they were not enriched

in specific gene ontology categories (Supplementary Data 7) and the sites of homoeologous exchange did not correspond to the sites of homologous recombination identified in introgressed segments in Timor hybrid derivatives (Supplementary Fig. 61). Smaller exchanges corresponding to $ROH_H$ peaks involving one or a few individual genomic windows[43] visible in Fig. 4 may have gone undetected due to

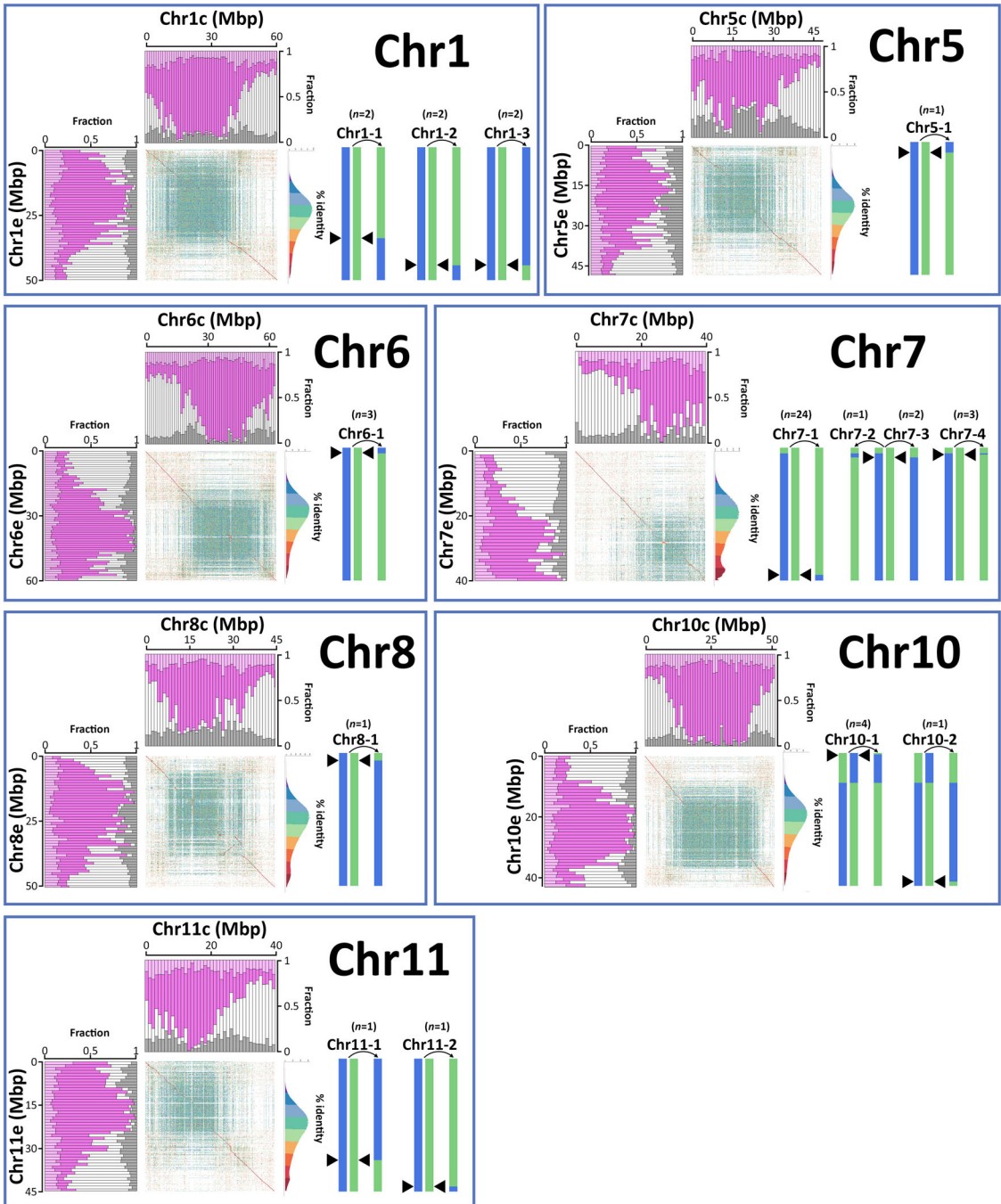

**Fig. 5 | Graphical representation of large homoeologous exchanges in *C. arabica*.** Blue and green vertical ideograms represent native canephora and eugenioides homoeologous copies, respectively, and the chromosome(s) resulting from homoeologous exchange found in *C. arabica* germplasm (> 200 Kb). Each event of homoeologous exchange is identified by a number as in Supplementary Data 6 and *n* indicates the number of accessions in which the event was detected. The black arrowheads indicate the approximate location of the exchange. Ideograms are not drawn to scale. The exact lengths are reported in Supplementary Table 2 and Supplementary Data 5-6. In the sequence identity plot, dots represents sequence alignments with >70% of identity between non-overlapping 2 Kb windows. The color of each dot represents the % of sequence identity. Bar plots illustrate structural variation between homoeologous chromosomes, showing the fraction of nucleotides shared between (white and gray) or private to (pink and magenta) the homoeologs. These categories are further sorted into the fraction of nucleotides in annotated transposable elements (gray and magenta) and in non-repetitive DNA (white). The pink stack includes low-copy DNA as well as other DNA tracts that are not annotated as transposable elements outside of collinear regions. The axis indicating the chromosomal coordinates in million base pairs (Mbp) of each homoeolog refers to both the bar plot and the sequence identity plot. Source data are provided as a Source Data file and in Supplementary Data 6.

the sensitivity limits of our analytical pipeline that were defined by the rather low coverage of most accessions. The vast majority of the events involved accessions from Ethiopia, including landrace varieties and accessions representing garden-based and forest-based coffee production systems. The only case of an exchange identified in a variety belonging to the Bourbon/Typica germplasm of Yemeni origin (Supplementary Figs. 59-60) was in the Costarica-1 accession where we detected a putative very large chimeric exchange event on chromosome 1. Based on the identification of polymorphic homoeologous exchanges in different plants of 'Bourbon', we searched for similar polymorphisms between different accessions of other widely known cultivars to address the question of whether what happened in

'Bourbon' was a rare case or a common situation. To this end, we resequenced two accessions of 'Geisha' (Supplementary Table 10) and compared them with one another as well as with a third and publicly available accession of 'Geisha' (Supplementary Data 1). The very large chimeric exchange event on chromosome 1 that we found in Costarica-1 appears to coincide with the same event present in heterozygous condition in the 1-Geisha accession (Supplementary Data 5) corresponding to the Ethiopian accession T.02722 that we sourced from CATIE (Supplementary Data 3). This event is not present either in the publicly available 'Geisha' (Supplementary Data 1) or in the accession we sourced from World Coffee Research (Supplementary Table 10), confirming that homoeologous exchanges in the Arabica germplasm are, at the same time, polymorphic between accessions that are identified with the same name and shared between accessions that are identified with different names.

A large number of events involved chromosome 7, most occurring at the opposite end from which all 4 homoeologs carry an eugenoides-derived chromosomal region and thus are in an autopolyploid condition. Events were also detected immediately downstream of the end that is in a autopolyploid condition, but those occurring in the very terminal portion of the chromosome where the exchange is already present and fixed in all accessions would have been undetectable because they would not have led to changes either in DOC or $ROH_H$. The formation of either tetravalents or bivalents involving homoeologous chromosomes at meiosis, which has been shown to occur in *C. arabica* by independent experiments[18,19], may be favored by this condition and may explain the high frequency of homoeologous exchanges observed for this chromosome. Overall, all of these events across different chromosomes suggest that homoeologous pairing in *C. arabica* is far from being completely or severely suppressed[47]. We tried to cluster events with similar chromosomal coordinates to indicate shared events among accessions (Fig. 5, Supplementary Fig. 62 and Supplementary Table 9), even though the low resolution of the analysis due to the large window size used to compensate for the low read coverage does not warrant their identity by descent. With this caveat, a similar exchange event to the one we detected in heterozygous condition in the Bourbon specimen used for ONT sequencing was also detected in heterozygous state in an Ethiopian accession used in the forest-based coffee production system (DASF2), while an apparently reciprocal event was identified, again in heterozygous condition, in two other Ethiopian accessions belonging to the same group (GSSF5 and YASF2, Supplementary Data 1). All 3 accessions are quite distantly related to each other and very distant from the Bourbon/Typica group based on the phylogenetic tree built on genome-wide SNPs (Supplementary Fig. 45). As many as 25 out of the 45 events we detected were in heterozygous condition and this may indicate a very recent origin as a consequence of the very frequent occurrence of exchanges between homoeologs. The observation of only one fixed homoeologous exchange event in the arabica germplasm among those that surpassed the size threshold we used for χ-scan analysis (the event at the top of chromosome 7) and of multiple other events that are still polymorphic in the germplasm lends additional support to the hypothesis that the origin of the species is recent.

A DOC level different from that expected for an allotetraploid when considering both pairs of homeologous chromosomes, in combination with $ROH_H$ allowed us to also infer the occurrence of large events that change the total copy number of chromosome segments as a consequence of either duplications or deletions (Supplementary Fig. 58 and Supplementary Data 4). A single duplication event involving a 1.5 Mbp of sequence at the top of Chr4e was detected in a single accession in heterozygous condition. We could not, however, determine the current location of the duplicated segment. Two large deletions were detected, each in a single accession, one involving a very large segment of approximately 14.2 Mbp at the top of Chr3c and the other a segment of 2.2 Mbp at the bottom of Chr7e. While the first one

was inferred to be in heterozygous condition, the second one was in homozygous condition. All three events were terminal and not interstitial along the chromosomes. While polyploid individuals may tolerate the occurrence of deletions more easily than diploid ones, especially when in homozygous condition, the observation of a much more limited number of deletion events in comparison to the number of homoeologous exchanges when combined also with the very rare occurrence of monosomies may point to the fact that these events are also less tolerated in an allotetraploid species such as *C. arabica*. On the other hand, the observed differences may simply derive from a difference in the frequency of occurrence of the different types of chromosomal mutations. Even in the case of deletions and duplications, we have to stress again that smaller events corresponding to $ROH_H$ peaks involving one or a few individual genomic windows may have gone undetected due to the sensitivity limits of our analytical pipeline.

While none of the different chromosomal aberrations we have identified is introducing new sequence variants, they are all capable of altering gene dosage between homoeologs and also total gene dosage in the case of aneuploidies, deletions and duplications. These changes in gene copy number may result in expression differences and this in turn may result in significant phenotypic variation, as shown by aneuploid somaclonal variants that frequently originate from tissue culture and somatic embryogenesis[20]. The genomic rearrangements may therefore represent an important source of genetic diversity that is continuously replenishing the very limited nucleotide diversity available within the species.

The development of a chromosome-level assembly of *C. arabica* using long read technology has allowed us to perform a detailed comparative analysis of the structure and evolution of the two parental genomes that form this species. Within each pair of homoeologous chromosomes, we observed the presence of two different chromatin compartments that corresponded to regions with markedly different structural features and evolutionary history: A chromatin compartments show high gene density and low sequence divergence while pericentromeric B chromatin compartments show low gene density and high sequence divergence due to recent TE insertions. Expression differences between homoeologous genes are higher for genes in B chromatin compartments than in A chromatin compartments, even though their number is much smaller. When we considered the genes putatively involved in caffeine biosynthesis we detected both variation in copy number as well as expression differences between the homoeologs. The expression differences were always in the direction of higher expression in the canephora homoeologs than in the eugenioides ones, as previously observed[35]. Additional experiments are needed to identify the elusive centromeric regions in *C. arabica*, such as chromatin immunoprecipitation sequencing to identify sequence arrays that support centromere protein A occupancy[2].

As to the events that took place after the polypoidization, in addition to more precisely defining three events of non-reciprocal chromosomal exchanges between homoeologs that had previously been described, we identified one event of reciprocal exchange. Different lines of evidence point to the fact that these four events seem to be common to all accessions analysed and are therefore likely to have occurred very early on after the formation of *C. arabica*. More importantly, we identified in a large set of accessions many additional putative chromosomal aberrations, consisting of different types of aneuploidies, deletions and duplications, as well as non-reciprocal chromosomal exchanges between homoeologs that were private to single or groups of accessions, and found both in homozygous as well as heterozygous condition. These events appear to be recent and could represent a mechanism by which genetic variation is created in a species that is otherwise confirmed to have very low levels of nucleotide diversity across its genomes. The analysis of single nucleotide variation based on whole-genome resequencing data in the large set of

accessions revealed the presence of putatively introgressed genomic segments from *C. canephora* in individuals in which these introgressions were expected based on their known pedigrees as well as in a small number of individuals that were assumed to be representative of *C. arabica* cultivars, landraces and ecotypes. The comparison of these introgressed segments with those present in derivatives of the Timor hybrid, a plant resulting from a cross between a *C. arabica* accession and a *C. canephora* accession and carrying genes providing resistance to coffee leaf rust, revealed their common origin. The introgression events that were detected appear to result from recent contamination events of the present-day seed pool and may pose a threat to the genetic purity if new Arabica plantations were to be established using sexually propagated material from these sources.

## Methods

### DNA sequencing and de novo assembly using ONT and Hi-C reads

WGS libraries were prepared using 2–4 µg of high molecular weight genomic DNA of 'Bourbon' and the 1D DNA Ligation Sequencing kit SQK-LSK109 (Oxford Nanopore Technologies, Oxford, UK), following manufacturer's instructions with the following modifications: incubation at 20 °C, extended 'DNA repair & End-prep' to 60 min, purification of adapter ligation reactions using an higher concentration (0.45X) of Agencourt AMPure XP beads (Beckman Coulter, Brea, CA). DNA quality was checked using Qubit 2.0 Fluorometer (Invitrogen, Carlsbad, CA). The libraries were sequenced using a PromethION sequencer (Oxford Nanopore Technologies, Oxford, UK), generating 121 Gbp with an average read length of 32 Kb. ONT reads were corrected and assembled with Canu[48] with default parameters but raw ErrorRate=0.4, correctedErrorRate=0.144, minReadLength=10000, minOverlapLength=3000, ovlMerDistinct=0.975, corMhapSensitivity=high, saveReads=True, "batOptions = -dg 3 -db 3 -dr 1 -ca 500 -cp 50". Contigs were polished using 29.5 Gbp of Illumina reads (BioProject PRJNA554647, SRR9822011-15) with a 20-fold iteration of read alignment to the reference sequence using BWA-MEM[49] and base correction performed using Pilon[50]. Hi-C libraries were prepared using the Arima Genome-Wide HiC+ kit (Arima Genomics, Carlsbad, CA) following manufacturer's instructions and sequenced using a NovaSeq 6000 instrument (Illumina, San Diego, CA). Hi-C reads were aligned with the reference genome using BWA-MEM[49] with the option parameter -5SP and filtered using samblaster, the command samtools view -F 2316 and matlock bamfilt with default parameters (https://phasegenomics.github.io/2019/09/19/hic-alignment-and-qc.html). Identification of chimeric contigs and scaffolding was performed with SALSA[51] with default parameters but -e GATC,GANTC. Genome-wide contact maps were generated using Juicer and plotted using Juicebox[52]. Chromosome pseudomolecules were aligned with previous *C. arabica* genome assemblies (GCF_003713255.1) and oriented and coded accordingly.

### Validation of sequence accuracy and chromosomal reciprocal exchanges

BAC clone sequences were retrieved from the BioProject PRJNA554647[6] and assembled using ABySS[53]. Scaffolds longer than 10 Kb that had been obtained from pooled sequencing of 384 random BAC clones were mapped on the reference using MUMmer[54]. The intervals of sequence alignment were retained with a minimum threshold of 95% sequence identity. Partial contig alignments of repetitive sequences were filtered out by retaining only adjacent matches within a 200 Kb range that collectively accounted for >30% of the scaffold length. We excluded 8 chimeric scaffolds carrying two portions that mapped onto two different chromosomes likely originating from the procedure used for the scaffolding of BAC contigs[6] and 5 BAC contigs that matched multiple reference regions with similar levels of sequence identity likely originating from repetitive DNA. A total of 1.5 Mbp were used for estimating the sequence error rate. The primers pairs used for PCR-based validation assays of chromosomal reciprocal exchanges are reported in Supplementary Table 11.

### Gene and TE prediction

Gene prediction was performed using complementary evidence provided by alignments of plant proteins obtained from *C. arabica* (GenBank Assembly Accession GCA_003713225.1) and from Uniprot Gentianales, OrthoDB and SwissProt databases, by RNA read alignments performed as described below, and by ab initio gene prediction using SNAP[55], Glimmer[56], BRAKER2[57] and Geneid[58] software. The final gene models were generated using EvidenceModeler and PASA[59]. Gene Ontology IDs were assigned using Pannzer2[60] and grouped into coarse-grained terms using the goslim_plant database. Gene Ontology enrichment analysis was performed using the R package topGO. Intact TEs were identified and classified using EDTA[61]. Repetitive DNA was masked using RepeatMasker[62] and the TE library generated by EDTA. Chromovirus-domain containing Gypsy and Athila sequences were extracted from *C. canephora* using the annotation of intact LTR elements of Zhou and coworkers[63] and the library was used for masking the *C. arabica* assembly with RepeatMasker[62].

### Genome segmentation and genomic and Hi-C windows analyses

Chromosome pseudomolecules were segmented into 4,467 non-overlapping genomic windows of variable size, containing 100 Kb of non-repetitive DNA. This segmentation in windows of variable length and fixed low-copy DNA amount were used for analyses of SNP frequency, nucleotide diversity, gene and TE densities and *C. canephora* introgressions. Chromosome-scale visualization of tandem repeat structures with identity heatmaps were generated with StainedGlass[64] using 2 Kb genomic windows. Synteny plots were obtained with SyRI[65]. Principal component values on aligned Hi-C reads were produced using the HOMER utility runHiCpca.pl[66]. PCA was applied to full chromosome distance-normalized interaction matrices at 50 Kb resolution. Each region along the chromosome represents a dimension in the analysis. It has been previously shown that the first eigenvector (PC1) broadly reflects clustering by chromatin type, i.e. heterochromatin or euchromatin and allows for identification of genome compartments[67]. A and B compartments were classified according to the sign of the first component (PC1) values, where positive values identified A compartments and negative values identified B compartments. Since the PC1 eigenvectors sign may be inconsistent across chromosomes, a manual correction of the signs was carried by direct inspection of the contact map.

### Reference-based DNA and RNA read alignments

Short DNA and RNA reads were aligned with a modified version of the reference genome in which the nearly identical copy of the homoeologous replacement on the displaced homoeolog has been masked in order to allow uniquely read mapping. RNA reads were retrieved from BioProject PRJNA554647 and aligned using STAR[68]. Gene expression data were generated using StringTie and read counts were normalized as Transcripts Per Million (TPM). Differences between A and B compartments in expression levels and in ratios of expression between homoeologs were tested using a two-sided Wilcoxon test.

### Sequence variation and homoeologous copy number variation analyses

Raw data were retrieved from public repositories (Supplementary Data 1 and 3). DNA reads were aligned with the reference genome using BWA-MEM[49] with default parameters. Uniquely mapping DNA reads were retained with a mapping quality >10. In the case of WGS data, raw variants were called using the UnifiedGenotyper tool in GATK[69] with 0.01 heterozygosity parameter. Genotypes were called with a minimum coverage of 10 reads. Minimum coverage was reduced to 5 reads for low

coverage sequencing data downloaded from public repositories. Heterozygous genotypes were called with a reference/alternative read coverage ratio between 0.15 and 0.85. Homozygous reference genotypes were called with a reference/alternative read coverage ratio ≤0.1. Homozygous alternative genotypes were called with a reference/alternative read coverage ratio ≥0.9. We used the calls obtained from the self-alignment of Bourbon Illumina reads (30× genome coverage) to filter out false variant sites arising from misalignment or sequence inaccuracy in the reference genome assembly. Phylogenetic trees in *C. arabica* were constructed using vcf-kit. In the case of GBS data, raw variants were called using Stacks[70]. Genotypes were called with a minimum coverage of 10 reads. Heterozygous genotypes were called with a reference/alternative read coverage ratio between 0.25 and 0.75. Homozygous genotypes were called as described for WGS. Variant sites were retained if informative in >50% of the individuals. For the detection of chromosomal aberrations and homoeologous exchanges, alignments of WGS reads and SNP calling were perfomed with the same software and parameters as described above except that DNA reads were aligned to each subgenome of the reference, separately. Homoeologous SNPs were used to identify deviations from the variant frequency that is expected under the normal condition of balanced homoeologous copy number using the the software χ-scan[71] in sliding windows of variable size containing 500 variant sites with an overlap of 250 variant sites. Depth of Coverage (DOC) was calculated using the command *genomecov* in *bedtools* in 4,467 non-overlapping genomic windows of variable size, containing 100 Kb of non-repetitive DNA, and normalized to the DOC in 'Bourbon'. Details on $ROH_H$ and DOC thresholds used to determine copy number variation are given as Supplementary Method 7.

### Reporting summary

Further information on research design is available in the Nature Portfolio Reporting Summary linked to this article.

## Data availability

The data generated in this study have been deposited in the NCBI database under the following BioProject numbers: raw sequences and the genome assembly of 'Bourbon' PRJNA944143, raw sequences of 4 accessions (Supplementary Table 10) PRJNA1001613 and PRJNA1001614. The following genome features of the Bourbon assembly are graphically available using the genome browser at https://coffea.appliedgenomics. org/ and accessible upon registration: gene predictions supported by evidence of RNA read alignments, repeat annotation, k-mers, synteny and collinearity between homoeologs. Raw sequence data for genetic diversity analysis were obtained from the BioProjects PRJNA505204, PRJNA790687, PRJNA554647, PRJNA497891. Large data sets are deposited in the figshare repository[43] [https://figshare.com/articles/figure/b_ A_chromosome-scale_assembly_reveals_chromosomal_aberrations_and_ exchanges_generating_genetic_diversity_in_b_b_i_Coffea_arabica_i_b_b_ germplasm_b_/23821881] with the DOI [https://doi.org/10.6084/m9. figshare.23821881]. They include Reduction Of Homoeologous Heterozygosity ($ROH_H$) and Depth of Coverage (DOC) analysis in *C. arabica* with data organized according to individual genotype as well as to individual chromosome, including simulations of read coverage variation; introgression analysis in the GBS diversity panel; genome annotation data (gene prediction and repeat annotation in GFF3 File Format; gene annotation in txt format). Source data are provided with this paper.

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

## Acknowledgements

This study has been funded by illycaffè SpA, Lavazza Group and Istituto di Genomica Applicata. We thank Irena Jurman, Nicoletta Felice, Alessandro Spadotto, Eleonora Paparelli, Giorgia Di Santolo and Luca Turello for technical assistance and Rachel Schwope for English language editing. We thank Area Science Park for hosting the assembly computation at the ORFEO Data Center.

## Author contributions

S.S. and D.S. assembled the reference genome; G.M. and M.L. carried out genome structure analysis and genetic analysis and plotted the graphs; N.V. performed gene prediction and gene functional annotation; M.V. generated the genome browser; L.D.T., M.R.R., L.N., G.P., F.S.L., M.C., J.C.B.M.T., L.T. participated in the conception of the study; G.D.G. and M.M. interpreted the data, assembled the figures and wrote the manuscript. All authors revised the final version of the manuscript.

## Competing interests

All authors declare that there is no outcome reporting bias that may support the interests of the employers and financial sponsors. Selection bias was excluded by the use of all publicly available sequencing data for genetic diversity analysis. L.D.T., M.R.R., L.N., G.P., F.S.L., M.C. are employees of coffee roasting companies with R&D responsibilities. Other authors claim no competing interests.
