## [Peer Review File · Nature Communications]

A chromosome-scale assembly reveals chromosomal aberrations and exchanges generating genetic diversity in *Coffea arabica* germplasmReviewers' Comments:

Reviewer #1:

Remarks to the Author:

The manuscript reports an improved arabica coffee genome. The key findings are the presence of introgressed regions from robusta. This suggests that human and wild selection has already achieved greater stress tolerance in arabica using robusta genes as an objective of many arabica genetic improvement efforts.

The work is carefully conducted and the conclusions are sound, delivering important results. The genome was produced by ONT, resulting in a less contiguous genome than that expected if high coverage with other sequencing technology had been performed. Haplotype resolution was not reported due to the methodology adopted. The sequence was also polished with Illumina reads, something that is to be avoided with highly accurate sequences due to the risk of removing real sequence variation, especially in repeated sequences. This work could be repeated with the information provided, but a better genome may be possible with high coverage (eg 200X) with HiFi reads.

Reviewer #2:

Remarks to the Author:

In this manuscript, the authors report a genome assembly and its qualitative evaluation, of the allotetraploid species, *Coffea arabica*. In addition to the classical parameters for the evaluation of genomic assemblies, aspects more specific to the species studied are addressed, such as the relationship between gene expression and chromosomal location, the synteny between the two subgenomes, or the expression of N-methyltransferase genes involved in caffeine biosynthesis.

Overall, the manuscript suffers from a lack of well-identified scientific questions, which reduces its appeal to most Nat-com readers. Its main outcome is the establishment of a new genomic resource for the *C. arabica* species. This genomic assembly appears to be of better quality than previously published assemblies (including by the same team for the same genotype). Provided that the generated data are effectively and fully available, this is of significant interest to the scientific community. In contrast, the additional analyses performed are not very significant for the research areas involved. Constitutive and functional differences between the heterochromatin and euchromatin compartments is a basic feature of eukaryotic genomes. The existence of a strong synteny between the genomes of coffee species in general and the progenitor species and subgenomes of *C. arabica* in particular has already been demonstrated many times (e. g. doi.org/10.1038/s41598-021-87419-0). The expression of homeologous genes in *C. arabica* has been studied in detail in several papers (e.g. [doi: 10.1111/nph.12371](https://doi.org/10.1111/nph.12371)); recently a detailed analysis of the expression of genes involved in biosynthetic pathways such as caffeine has been reported (doi.org/10.1093/aob/mcac041). The analyses presented in the present manuscript are therefore not very original and are handicapped by the experimental design (apparently only two replicates). Furthermore, the authors point to the presence of non-reciprocal intergenomic exchanges, in particular on chromosomes 2c, 7c and 10c. These events have already been studied and identified ([doi: 10.1534/g3.116.030858](https://doi.org/10.1534/g3.116.030858)). In the present manuscript, with the availability of a quality genomic sequence, one would expect a more detailed analysis of the mechanisms behind these events and their genetic consequences.

Finally, the authors give an important place (title as well as in the main text) to the supposed identification of recent introgressions of *canephora* in several traditional arabica varieties (African germplasm). These introgressions would be identical to those observed in the genetic material derived from the Timor hybrid (the origin of many so-called "modern" varieties). The authors interpreted this observation as the mark of creeping introgressions with a strong selective advantage. This result and this interpretation are, in my opinion, very doubtful. Indeed, this result goes against previous diversity analyses for this type of plant material, which are numerous and have involved several teams, including the authors of the present work. Moreover, it is difficult to conceive the possibility of several parallel

identical introgression events (same single *canephora* parent; transmission and selection of the same chromosomal fragments). Finally, this scenario is not compatible with the known chronology of events. The Timor hybrid was identified in a field of *C. arabica* (planted in 1927) on the island of Timor (Bettencourt 1973). The Timor hybrid is thought to be the result of a spontaneous interspecific cross between *C. arabica* and a genotype of *C. canephora* originating from the Congo Basin and introduced in Timor at the beginning of the 20th century. Although it was transferred to Portugal in 1957, worldwide dissemination of selected descendants of the Timor hybrid did not begin until the 1970s. In addition, traditional varieties (and reported as introgressed in the present work) such as Kent (in India, see Srinivasan & Narasimhaswamy, 1940) and SL34 (in Kenya, see Melville, 1946) were developed in the 1930s and 1940s, so their introgression by a descendant of the Timor hybrid seems hardly conceivable. Furthermore, the present study is based on the analysis of data from the sequencing of plant material from "second hand" collections (for example, a collection installed in China with material from other collections). The genetic conformity of the plant material can therefore be questioned, especially since these introductions into the collection were probably made by seed and not by cuttings. The often heterozygous state of the detected introgressions is also an indicator of possible crosses. In all cases, verification with certified true-to-type plant material would be essential.

Reviewer #3:

Remarks to the Author:

The article describes a high continuity assembly of the allotetraploid *Coffea arabica*, that was used for various analyses including the description of two chromatin compartments, the identification of chromosomal rearrangements between the two subgenomes, the comparison of N methyltransferases (involved in caffeine biosynthesis) between the two subgenomes, and the identification of pervasive *Robusta* introgressions in the *Arabica* cultivars.

The analyses performed on this assembly provide interesting findings about *C. arabica*. However, it is a bit excessive to qualify the assembly as "chromosome scale", since on average, each chromosome is made of 7.95 contigs (175 contigs for 22 chromosomes) and only one chromosome (7c) is made of one contig. The authors did not provide the number of scaffolds per chromosome, but data in table S1 suggest that they are similar. Table S2 should also display the number of scaffolds for each chromosome. The authors should discuss on how they define a chromosome level assembly and why they view this assembly as a chromosome-scale one.

Additionally, it is not clear whether this new assembly allows to raise more conclusions than the one described in lines 63-68 (Introduction) or whether the same analyses could have led to similar conclusions on the previous assembly. The authors should mention in detail which of the discoveries described in the current article were allowed by the new assembly (that were not possible with previous ones). If few, the article should be rephrased with an emphasis on the analyses rather than on the description of the "chromosome-scale" assembly, including the title.

Another major concern is the fact that few numerical data are provided: results are mostly displayed as figures, but the readers don't have access to the actual data that were used to build the figures and to perform the analyses. At the moment, the information provided in the manuscript is not detailed enough to allow to reproduce the findings. In the same way, methods are not sufficiently described (see below for more details): the text does not explain the approach used to obtain results but goes straight to the results. The reader should at least be able to find paragraphs corresponding to the text in the Methods section.

Most figures (main and supplementary) are not well defined when printed and very hard to read in some cases (font too small).

the resolution or size should be improved. When printed, the text is very small and hard to read. (it is also the case for Figure 2, and most supplementary figures)

Finally, in addition to the fact that the supplementary tables do not contain enough data, and that the methods are not detailed enough, the data availability section lacks the link to the genome browser. Moreover, the bioproject PRNA944143 does not seem to be publicly available yet, which complicates the assessment of the results presented in the manuscript.

Below are more specific comments:

Line 105: "This assembly shows more consistency in order and orientation with the assemblies of the diploid progenitors than previous *C. arabica* assemblies (Figure S1)": Figure S1 does not really allow to compare the new assembly with the previous one regarding contiguity with the diploid progenitors: the comparison would be much easier if the progenitors were in the middle, and the two *C. arabica* assemblies on each side. Moreover, it would be very valuable to have a metric to show the better contiguity, rather than only a visual inspection.

Figure 1 : Would it be possible to add information about A and B compartments as in Figure S4 ?

Line 153 : "We performed a principal component analysis..." Here, although the Methods section describes the software used (Homer runHiCpca.pl), a more detailed description of the PCA should be provided. At least explain what variables were used in the PCA : was it applied to the distance-normalized HiC interaction matrix? What are the most contributive variables in PC1 ? What can explain that PC1 discriminates between euchromatin and heterochromatin?

Lines 171-181 : the numbers and % are hard to follow. It would be useful to have the numeric values in a table (not only displayed in Figure S6 – the same goes for all data described in the manuscript-), at least for the numbers cited in the text such as: "444Mb" of sequences where the two subgenomes are collinear, and "519 Mb" of "remaining parts" highly enriched in structural variants. Should not the different parts sum up to 1,099 Mb (size of assembled chromosomes)?

Lines 204-238 : the study of N methyl-transferases is interesting but a bit too descriptive. It would be worthwhile to discuss the possible implications of sequence polymorphism (and tandem repeat copy number variation) as well as expression level differences between eugenioides and canophora-derived homeologs on caffeine biosynthesis.

Chromosomal rearrangements between subgenomes (lines 314-334):

The discovery of a recent homeologous exchange in one Bourbon specimen is very interesting. Since the authors used resequencing data from a large set of accessions for the genetic diversity analysis, would it be feasible to look for other recent homeologous exchanges in all these specimens, using the depth of mapping of reads on the reference genome (and maybe identify other specimens sharing the event identified in the Bourbon specimen).

Centromeric regions, lines 275-283: It is not clear whether the authors make the hypothesis that the 27 unanchored scaffolds containing arrays of the monomer identified in chr7c correspond to centromeric regions for the other chromosomes. If yes, the hypothesis should be formulated in a more straightforward way. Since the centromeres could not be resolved in most chromosomes, it might be interesting to discuss possible additional technologies that could help improve the assembly in these regions (optical maps?) (maybe in the Conclusions section).

Finally, the methods section needs to be extended: find below a non-exhaustive list of the paragraphs that should be added or completed.

- As already stated, details on the PCA analysis performed to identify genomic compartments should be provided, as well as the input matrix for PCA as an additional table.
- Statistical tests performed to show significant differences between A and B compartments: some information stand in the legend of the supplementary figure S5, but it would be useful to have it in the

dedicated methods section

- Dating of LTR retrotransposons insertions : how was it performed?
- Comparison of transcription levels between A and B compartments (Fig S8,S9)/ Section of methods only describes how TPM were calculated. Need of more details on the statistical analysis.
- Methods to identify homeologous exchanges. Line 287 "we precisely mapped the location of three subtelomeric events": How was the analysis performed ?

REVIEWER COMMENTS

Reviewer #1 (Remarks to the Author):

The manuscript reports an improved arabica coffee genome. The key findings are the presence of introgressed regions from robusta. This suggests that human and wild selection has already achieved greater stress tolerance in arabica using robusta genes as an objective of many arabica genetic improvement efforts. The work is carefully conducted and the conclusions are sound, delivering important results.

The genome was produced by ONT, resulting in a less contiguous genome than that expected if high coverage with other sequencing technology had been performed. Haplotype resolution was not reported due to the methodology adopted. The sequence was also polished with Illumina reads, something that is to be avoided with highly accurate sequences due to the risk of removing real sequence variation, especially in repeated sequences. This work could be repeated with the information provided, but a better genome may be possible with high coverage (eg 200X) with HiFi reads.

The risk of removing real sequence variation in repeated sequences using Illumina reads for polishing ONT reads is not real because only uniquely mapping reads are used for the polishing and by definition the repeated sequences will correspond to regions where reads do not map uniquely, so the polishing does not take place in those regions. As to the possibility of obtaining a better quality genome using HiFi reads, this is definitely true but we feel that the assembly we are providing represents an improvement over the already available ones (see responses to reviewer 3) and is more than adequate in order to perform analysis on genetic variation within the species *Coffea arabica* as shown by our new analyses on chromosomal aberrations.

Reviewer #2 (Remarks to the Author):

In this manuscript, the authors report a genome assembly and its qualitative evaluation, of the allotetraploid species, *Coffea arabica*. In addition to the classical parameters for the evaluation of genomic assemblies, aspects more specific to the species studied are addressed, such as the relationship between gene expression and chromosomal location, the synteny between the two subgenomes or the expression of N-methyltransferase genes involved in caffeine biosynthesis.

Overall, the manuscript suffers from a lack of well-identified scientific questions, which reduces its appeal to most Nat-com readers. Its main outcome is the establishment of a new genomic resource for the *C. arabica* species. This genomic assembly appears to be of better quality than previously published assemblies (including by the same team for the same genotype).

The scientific question we have focused on, especially in the revised version of the manuscript, is the identification of previously undetected sources of genetic variation that we have identified as aneuploidy, homoeologous exchanges and introgressions from *Coffea canephora*. We feel this question to be particularly important for a species where genetic variation, as estimated as nucleotide diversity, is extremely low.

Provided that the generated data are effectively and fully available, this is of significant interest to the scientific community.

All data will be made available to the community as soon as the paper is accepted, as per Nature Communications policy.

In contrast, the additional analyses performed are not very significant for the research areas involved. Constitutive and functional differences between the heterochromatin and euchromatin compartments is a basic feature of eukaryotic genomes.

This is indeed true but this information was not yet available for coffee and we have related it also to a number of structural and functional properties such as gene density, interhomeolog expression differences, structural variation due to transposable element insertions etc.

The existence of a strong synteny between the genomes of coffee species in general and the progenitor species and subgenomes of *C. arabica* in particular has already been demonstrated many times (e. g. doi.org/10.1038/s41598-021-87419-0).

We fully agree with the reviewer that information was already available on this topic but we do not feel this is a major focus of our manuscript. On the other hand, we feel that the comparison of syntenic relationships between subgenomes in the context of the genome organization into eu- and heterochromatic compartments (A and B domains, respectively) was not previously described and highlighted the presence of large genomic regions where the recent movement of transposable elements has greatly reduced the proportion of shared sequence between subgenomes. Additionally, the genome browser we are going to make available to the community will show to the users the individual syntenic intervals, allowing for a quick comparison between subgenomes.

The expression of homeologous genes in *C. arabica* has been studied in detail in several papers (e.g. doi: 10.1111/nph.12371); recently a detailed analysis of the expression of genes involved in biosynthetic pathways such as caffeine has been reported (doi.org/10.1093/aob/mcac041). The analyses presented in the present manuscript are therefore not very original and are handicapped by the experimental design (apparently only two replicates).

We fully agree with the reviewer that information was already available on this topic and added the suggested citations but our analysis of expression differences between homeologous genes was done to relate this aspect to chromatin domain organization and transposon related structural differences between subgenomes. As to the caffeine biosynthetic genes, the difference in expression between homeologs are also discussed in relation to structural differences existing for both genic as well as intergenic regions as well as gene duplications observed in the eugenioides derived subgenome for DXMT. As to the fault in experimental design due to the use of only two replicates, it has to be stressed that while biological replicates are very important when using absolute levels of gene expression in a comparison among treatments or among tissues in order to control for environmental sources of variation among replicates, they are much less relevant when comparing relative expression between two alleles in a gene (allele-specific expression analysis) or two homeologous genes such as in the present case or when comparing absolute levels between different sets of genes in the same sample since the environmental variation factors acting on the two genes or alleles or sets of genes under comparison are going to be the same because they (alleles, genes or sets of genes under comparison) are expressed in the same cellular environment (see these reviews for a discussion on sources of errors in Allele specific analysis of gene expression, which is equivalent to homeolog specific analysis that we performed here: <https://www.ncbi.nlm.nih.gov/pmc/articles/PMC7985293/>; <https://www.ncbi.nlm.nih.gov/pmc/articles/PMC7241832/>; <https://www.ncbi.nlm.nih.gov/pmc/articles/PMC4574606/>). Even when absolute levels of expression are shown such as in Supplementary Figures 8 and 11, these values are always used for comparisons within samples, such as between homeologs (Supplementary Figure S11) or between genomic compartments (Supplementary Figure S8) and could have been replaced by ratios of expression levels.

Furthermore, the authors point to the presence of non-reciprocal intergenomic exchanges, in particular on chromosomes 2c, 7c and 10c. These events have already been studied and identified (doi:

10.1534/g3.116.030858). In the present manuscript with the availability of a quality genomic sequence one would expect a more detailed analysis of the mechanisms behind these events and their genetic consequences.

We have added details on the molecular characterization of the already identified events in Bourbon, including the new one that is present in heterozygous condition. We have also added information on a large number of previously unidentified chromosomal aberration events present in the germplasm that was resequenced using Illumina sequencing. Due to the relatively low coverage and the short reads available for these accessions, the characterization was however at the genomic window level, and we did not attempt to go to identify individual breakpoints. We feel that this represents an important and novel piece of information since it extends the idea proposed in the Lashermes et al. 2016 (doi:10.1534/g3.116.030858) paper where most of the inter-genomic DNA exchanges appeared to be common to all varieties analysed. We have also shown that in addition to the inter-genomic DNA exchanges aneuploidies can also play a role in generating new variability within the *Coffea arabica* germplasm.

Finally, the authors give an important place (title as well as in the main text) to the supposed identification of recent introgressions of *canephora* in several traditional *arabica* varieties (African germplasm). These introgressions would be identical to those observed in the genetic material derived from the Timor hybrid (the origin of many so-called "modern" varieties). The authors interpreted this observation as the mark of creeping introgressions with a strong selective advantage. This result and this interpretation are, in my opinion, very doubtful. Indeed, this result goes against previous diversity analyses for this type of plant material which are numerous and have involved several teams including the authors of the present work. Moreover, it is difficult to conceive the possibility of several parallel identical introgression events (same single *canephora* parent; transmission and selection of the same chromosomal fragments). Finally, this scenario is not compatible with the known chronology of events. The Timor hybrid was identified in a field of *C. arabica* (planted in 1927) on the island of Timor (Bettencourt 1973). The Timor hybrid is thought to be the result of a spontaneous interspecific cross between *C. arabica* and a genotype of *C. canephora* originating from the Congo Basin and introduced in Timor at the beginning of the 20th century. Although it was transferred to Portugal in 1957, worldwide dissemination of selected descendants of the Timor hybrid did not begin until the 1970s. In addition, traditional varieties (and reported as introgressed in the present work) such as Kent (in India, see Srinivasan & Narasimhaswamy, 1940) and SL34 (in Kenya, see Melville, 1946) were developed in the 1930s and 1940s, so their introgression by a descendant of the Timor hybrid seems hardly conceivable.

We fully agree with the reviewer that the situation that we have described, and that the reviewer has recalled here, is not a condition that occurred at the origin of the mentioned cultivated varieties (i.e. in the generation of the seedling from which the lineage had been historically selected) but, as a matter of fact, sampling of present-day material that goes under those varietal names from gene banks or from plantations indicates that gene flow from HDT derivative is occurring. We think that this is a very important point with both commercial and scientific implications. On the one hand, it threatens the genetic purity of seed lots of *Arabica* varieties that may be used for establishing new plantations and, on the other hand, it confounds population genetics analyses as this material enters gene banks and its sequencing data populate public repositories, which we showed here has already occurred.

In the R1 version, we newly generated resequencing data from 4 additional accessions that support the above interpretation.

Even more importantly, the fact that one accession present in public databases is proving the occurrence of gene flow from *robusta* into the forest-based production system in Ethiopia should raise an alert on the survival of natural uncontaminated populations of *C. arabica*.

Furthermore, the present study is based on the analysis of data from the sequencing of plant material from "second hand" collections (for example, a collection installed in China with material from other collections). The genetic conformity of the plant material can therefore be questioned, especially since these introductions into the collection were probably made by seed and not by cuttings. The often heterozygous state of the detected introgressions is also an indicator of possible crosses. In all cases, verification with certified true-to-type plant material would be essential.

We fully agree with the reviewer on this point, which we pointed out in the closing paragraph of the discussion of the original version of the manuscript. In the R1 version, we used one accession introduced from CATIE (SL-28) to provide support to this interpretation.

Reviewer #3 (Remarks to the Author):

The article describes a high continuity assembly of the allotetraploid *Coffea arabica*, that was used for various analyses including the description of two chromatin compartments, the identification of chromosomal rearrangements between the two subgenomes, the comparison of N methyltransferases (involved in caffeine biosynthesis) between the two subgenomes, and the identification of pervasive *Robusta* introgressions in the *Arabica* cultivars.

The analyses performed on this assembly provide interesting findings about *C. arabica*. However, it is a bit excessive to qualify the assembly as "chromosome scale", since on average, each chromosome is made of 7.95 contigs (175 contigs for 22 chromosomes) and only one chromosome (7c) is made of one contig. The authors did not provide the number of scaffolds per chromosome, but data in table S1 suggest that they are similar. Table S2 should also display the number of scaffolds for each chromosome. The authors should discuss on how they define a chromosome level assembly and why they view this assembly as a chromosome-scale one.

We have made changes to Tables S1 and S2 to better reflect the complex procedure that was used to go from the *de novo* assembly contigs to software-generated scaffolds and finally to manually curated superscaffolds corresponding to the chromosomal pseudomolecules. We have also added details on the procedure used in the methods section of the Supplementary Information file. While this assembly is definitely not a telomere-to-telomere (T2T) assembly (except for chromosome 7c), which would imply that a single contig covers the entire chromosome, we feel it is definitely up to the standards of many other published assemblies that have been defined chromosome level assemblies (e.g. see Lin et al. 2021, *Genome Biology* Chromosome-level genome assembly of a regenerable maize inbred line A188 <https://genomebiology.biomedcentral.com/articles/10.1186/s13059-021-02396-x>).

It also fits the Chromosome Assembly definition given by NCBI and used by the Genome Reference Consortium (<https://www.ncbi.nlm.nih.gov/grc/help/definitions/>):

"a relatively complete pseudo-molecule assembled from smaller sequences (components) that represent a biological chromosome. Relatively complete implies that some gaps may still be present in the assembly, but independent measures suggest that most of the sequence is represented by sequenced bases. Completeness is submitter defined."

The definition of a chromosome-level assembly refers to the fact that most sequences are assigned to chromosomes and gaps are limited within those sequences.

Additionally, it is not clear whether this new assembly allows to raise more conclusions than the one

described in lines 63-68 (Introduction) or whether the same analyses could have led to similar conclusions on the previous assembly. The authors should mention in detail which of the discoveries described in the current article were allowed by the new assembly (that were not possible with previous ones). If few, the article should be rephrased with an emphasis on the analyses rather than on the description of the “chromosome-scale” assembly, including the title.

The new assembly represents, in our opinion, an improvement over even the Caturra assembly described in lines 63-68. The N50 of the contigs is considerably larger for our assembly than the other (10.2 vs 3.9 Mbp) indicating a greater completeness, the scaffolds assigned to the chromosomal pseudomolecules are in total considerably larger (1099 Mbp vs 990 Mbp), and finally there are chromosomal rearrangements missing in the Caturra assembly such as the non reciprocal exchange at the top of chromosome 7. Most importantly perhaps the Caturra assembly has never been described in a publication and is only available in GenBank. While most of the analyses could have probably been performed also with the Caturra assembly we feel that especially the newly added section on chromosomal rearrangements and the section on centromeric regions greatly benefitted from the more complete assembly.

We provided in the R1 version two comparative analyses of structural completeness and accuracy. They both suggest that using the new assembly as a reference genome for chromosome-scale analyses is more advantageous than using the existing Caturra assembly. Supplementary Table S3 now includes a chromosome-by chromosome comparison in term of pseudomolecule length and sequence contiguity. A newly added Supplementary Table S4 reports on the relative extent of inconsistencies in order and orientation between either the assembly of this paper or the Caturra assembly and publicly available assemblies of the diploid progenitors. Although the values reported in Supplementary Table S4 do not necessarily reflect, in absolute terms, the extent of assembly errors in the *C. arabica* reference genomes, because some of the inconsistencies may originate from inaccuracy in the diploid progenitor assemblies, the relative values suggest that the assembly of this paper is more consistent than the Caturra assembly with the assemblies of the diploid progenitors across the vast majority of the chromosomes.

Another major concern is the fact that few numerical data are provided: results are mostly displayed as figures, but the readers don't have access to the actual data that were used to build the figures and to perform the analyses. At the moment, the information provided in the manuscript is not detailed enough to allow to reproduce the findings. In the same way, methods are not sufficiently described (see below for more details): the text does not explain the approach used to obtain results but goes straight to the results. The reader should at least be able to find paragraphs corresponding to the text in the Methods section.

The actual data that are used to build each figure is made available to the readers as recommended by Nature Communications as an appended Source Data file. Additional Research Data consisting in large sets of output files are now deposited in the repository figshare. We agree on the lack of details for the methods and we added a detailed methods section in the Supplementary Information file.

Most figures (main and supplementary) are not well defined when printed and very hard to read in some cases (font too small).

the resolution or size should be improved. When printed, the text is very small and hard to read. (it is also the case for Figure 2, and most supplementary figures)

We are providing high resolution figures.

Finally, in addition to the fact that the supplementary tables do not contain enough data, and that the methods are not detailed enough, the data availability section lacks the link to the genome browser.

Moreover, the bioproject PRNA944143 does not seem to be publicly available yet, which complicates the assessment of the results presented in the manuscript.

The BioProject containing the reads

(<https://dataview.ncbi.nlm.nih.gov/object/PRJNA944143?reviewer=32b371n6m8bs15hjvf61h5e2e5>) has been made publicly available. For the time being the assembly multifasta file containing chromosomal pseudomolecules and unassigned scaffolds is also being made available in figshare (<https://figshare.com/s/d1a0fa56c18ec33fbdba>).

Below are more specific comments:

Line 105: “This assembly shows more consistency in order and orientation with the assemblies of the diploid progenitors than previous *C. arabica* assemblies (Figure S1)”: Figure S1 does not really allow to compare the new assembly with the previous one regarding contiguity with the diploid progenitors: the comparison would be much easier if the progenitors were in the middle, and the two *C. arabica* assemblies on each side. Moreover, it would be very valuable to have a metric to show the better contiguity, rather than only a visual inspection.

Supplementary Figure S1 has been changed according to the reviewer’s suggestion.

The consistency of each assembly with the diploid reference is now expressed by the metrics: 1- [(cumulative length of translocated or inverted segments)/total length of the arabica pseudomolecules], under the assumption that real structural variation and assembly errors in the diploid reference equally affects the estimation in each arabica assembly (reported in Supplementary Table S3).

Figure 1: Would it be possible to add information about A and B compartments as in Figure S4 ?

We made the change in panel C as suggested.

Line 153: “We performed a principal component analysis...” Here, although the Methods section describes the software used (Homer runHiCpca.pl), a more detailed description of the PCA should be provided. At least explain what variables were used in the PCA: was it applied to the distance-normalize HiC interaction matrix? What are the most contributive variables in PC1 ? What can explain that PC1 discriminates between euchromatin and heterochromatin?

PCA was applied to full chromosome distance-normalized interaction matrices at 50Kb resolution using the HOMER utility runHiCpca.pl. Each region along the chromosome represents a dimension in the analysis. It has been previously shown that the first eigenvector (PC1) broadly reflects clustering by chromatin type, i.e. heterochromatin or euchromatin and allows for identification of genome compartments. A and B compartments were classified according to the sign of the first component (PC1) values, where positive values identified A compartments and negative values identified B compartments. Since the PC1 eigenvectors sign may be inconsistent across chromosomes, a manual correction of the signs was carried by direct inspection of the contact map.

Lines 171-181: the numbers and % are hard to follow. It would be useful to have the numeric values in a

table (not only displayed in Figure S6 – the same goes for all data described in the manuscript-), at least for the numbers cited in the text such as: “444Mb” of sequences where the two subgenomes are collinear, and “519 Mb” of “remaining parts” highly enriched in structural variants. Should not the different parts sum up to 1,099 Mb (size of assembled chromosomes)?

Thanks for spotting this, we had made a mistake in computing percentages for the shared sequence that was corrected and we had not indicated a fraction of non shared sequence that does not match TEs. Now percentages sum to 100 and lengths sum to 1098 Mb. We corrected the text and added a Supplementary Table with these statistics (Supplementary Table S5).

Lines 204-238: the study of N methyl-transferases is interesting but a bit too descriptive. It would be worthwhile to discuss the possible implications of sequence polymorphism (and tandem repeat copy number variation) as well as expression level differences between eugenioides and canophora-derived homeologs on caffeine biosynthesis.

We modified as suggested.

Chromosomal rearrangements between subgenomes (lines 314-334):

The discovery of a recent homeologous exchange in one Bourbon specimen is very interesting. Since the authors used resequencing data from a large set of accessions for the genetic diversity analysis, would it be feasible to look for other recent homeologous exchanges in all these specimens, using the depth of mapping of reads on the reference genome (and maybe identify other specimens sharing the event identified in the Bourbon specimen).

We are very grateful for the suggestion, we have done this by using both the depth of coverage as well as a method we had previously developed based on allelic frequencies of heterozygous SNPs (using here SNPs between homeologs) and have discovered a large number of additional rearrangements. We feel this represents a very significant addition to our paper.

Centromeric regions, lines 275-283: It is not clear whether the authors make the hypothesis that the 27 unanchored scaffolds containing arrays of the monomer identified in chr7c correspond to centromeric regions for the other chromosomes. If yes, the hypothesis should be formulated in a more straightforward way. Since the centromeres could not be resolved in most chromosomes, it might be interesting to discuss possible additional technologies that could help improve the assembly in these regions (optical maps?) (maybe in the Conclusions section).

We modified as suggested.

Finally, the methods section needs to be extended: find below a non-exhaustive list of the paragraphs that should be added or completed.

- As already stated, details on the PCA analysis performed to identify genomic compartments should be provided, as well as the input matrix for PCA as an additional table.
- Statistical tests performed to show significant differences between A and B compartments: some information stand in the legend of the supplementary figure S5, but it would be useful to have it in the dedicated methods section
- Dating of LTR retrotransposons insertions: how was it performed?
- Comparison of transcription levels between A and B compartments (Fig S8,S9)/ Section of methods only describes how TPM were calculated. Need of more details on the statistical analysis.
- Methods to identify homeologous exchanges. Line 287 “we precisely mapped the location of three

subtelomeric events”: How was the analysis performed ?

...

These changes were made as suggested.

Reviewers' Comments:

Reviewer #1:

Remarks to the Author:

The genome sequence presented here is an improvement on that available but can easily be improved by the application of other methods. The value of this contribution is in the identification of the types of variation available in the coffee genome. The authors have explained their contribution well and recognized the limitations.

Reviewer #2:

Remarks to the Author:

This revised version of the manuscript presents significant improvements over the initial version. Most of the suggestions made by the various reviewers have been taken on board, which is very much to be appreciated. However, despite the efforts made, I feel that some comments have not been adequately taken into account and I am still not convinced by certain results, interpretations and conclusions.

As mentioned earlier, I have no doubts about the quality of this new genome sequence and its interest for the scientific community. The analysis of this resource carried out in the present manuscript as well as the availability and access to this new genome sequence through a dedicated website, are very much appreciated.

On the other hand, the analyses presented to study the diversification mechanisms present in *C. arabica* are less well mastered and not always correctly presented.

One of the limitations of the diversity analyses carried out is the quality of the genomic resources used (from public database). New sequencing has improved the situation, but unfortunately a large proportion of the genomic resources used are still of poor quality (and do not conform to their identification). Pointing out errors in collections, as well as the importance of having a quality approach when duplicating collections and putting them into culture, is interesting but constitutes a difficulty for the objectives of the present study.

Another weakness of the diversity analyses is linked to the ambiguity of the objectives of the work: is it to identify the mechanisms of diversification of the *C. arabica* species since its formation (around 600,000 years ago) or to better characterize the genetic diversity of plant material in collections as part of breeding programs. For instance, the authors mention in the title "identification of chromosomal aberrations and introgressions from *Coffea canephora* as important drivers of genetic variation in the species". While the contribution of "chromosomal aberrations" to the genetic variation identified in *C. arabica* is relatively well established, introgressions (under natural conditions) of *Coffea canephora* into the *C. arabica* genome have never yet been identified in the course of its evolution. *Senus stricto*, it cannot be said that introgression has been a driver of diversity in *C. arabica*. Of course, numerous artificial crosses with diploid species (including distant species such as *C. racemosa*, <https://doi.org/10.1023/A:1003427613071>) as part of breeding programs or spontaneous crosses with cultivars of diploid species (*C. canephora*, *C. liberica*) in anthropized environments (fields such as Timor Hybrid) have broadened the genetic diversity available for improving the species. For the latter, a relatively well-documented example is New Caledonia's spontaneous hybrids (<https://doi.org/10.1002/ece3.2055>; <https://doi.org/10.1139/G07-011>). These two objectives are different and must be clearly distinguished in the presentation and discussion of results.

In addition to specific comments, I will attempt to illustrate my points in the following paragraphs.

ABSTRACT

Lines 23-24: "a distal euchromatic one that has remained extraordinarily collinear between the two

subgenomes after the split of the two species". Difficult to understand. It would be preferable to speak of "diploid progenitor or parental genome" and not associate this observation with the allopolyploidisation step.

Line 26-27: "low baseline intraspecific nucleotide diversity that increases to interspecific levels in a part of the Arabica-like germplasm". Formally, this conclusion is difficult to grasp and highly debatable. For example, when a segment of the Ca subgenome (the "canephora" derived subgenome of *C. arabica*) is replaced by a C segment (from the "canephora" genotype progenitor of the Timor hybrid) or Ea segment (the "eugenioides" derived subgenome), can this really be considered as interspecific polymorphism?

Line 31: why "still"?

INTRODUCTION

Lines 77-80: OK but only concerns genetic improvement (not the evolution of the species per se), is very partial (many other crosses have been made in breeding programs) and very biased by the publicly available whole-genome resequencing data.

Lines 82-84: This remains a hypothesis (strong and not new) in the absence of results showing a direct link between phenotypic variation and chromosomal rearrangement. The term "chromosomal aberration" is not perfectly adapted. While it's well-suited to a case of aneuploidy, it's not appropriate for a simple interhomeologous segment exchange. More generally, it would be interesting and totally justified to introduce bibliography on characteristics such as meiotic behavior or reproduction modalities in *C. arabica* that condition the appearance and fate of the "chromosomal aberrations" detected. Contrary to the authors' assertions, the presence of anomalies during meiosis has been known for a long time (Krug CA and Mendes AJT, 1940). Meiosis control and the fate of non-balanced gametes have been little studied in *C. arabica*. Nevertheless, see <https://doi.org/10.1093/jhered/91.1.81> and <https://doi.org/10.1139/g04-048>. With regard to the possibilities of intergenomic recombination, I have identified <https://doi.org/10.1007/s001220100747>.

Lines 87-89: should be linked to the poor management of germplasm collections and variety distribution. The expression "Arabica-like germplasm" is a source of confusion. It must be clearly defined or avoided.

RESULTS AND DISCUSSION

Lines 200-243 (analyses of a group of genes involved in the caffeine biosynthetic) These aspects, as well mentioned in the manuscript, have already been the subject of publications. Authors should confine themselves to divergent/complementary points.

Lines 310-312 ("a reciprocal and symmetrical exchange, possibly originating from a mitotic crossing-over between homoeologous chromosomes that occurred in an interspecific hybrid diploid cell lineage before the polyploidization event"). Clarifications are needed. What are the arguments for a mitotic origin? In a diploid context (not linked to polyploidy), it would be preferable to speak of orthologous chromosomes. It should also be borne in mind that sequenced accessions of diploid species (*C. canephora*, *C. eugenioides*) are not the genotypes at the origin of the allopolyploidization event.

Figure 3B and corresponding text: I'm not comfortable with the way accessions are named. Knowing that a number of them aren't what they should be, a more neutral code would be better. For example, the current presentation tends to suggest that cultivars such as SL34 or Kent are introgressed when in fact they are not.

Figure 3C and corresponding text: Since the measurement of nucleotide diversity for *C. eugenioides* is

based on a single accession, it would be preferable not to present it. It might also be worth mentioning that *C. arabica* has a "fixed heterozygosity" (linked to its allopolyploid state) which represents significant nucleotide diversity.

Line 510 ("synthetic hexaploid wheat plants present an even higher frequency of aneuploidy. The identification of aneuploid *C. arabica* accessions is therefore not unexpected"): More than a comparison with bread wheat, the identification of aneuploids was expected on the basis of our knowledge of the meiotic behavior of *C. arabica* (see previous comment above). An aneuploid *C. arabica* plant (even dihaploid) can survive, but it is not possible to maintain it by sexual reproduction. Consequently, even if the phenomenon is frequent, its contribution to the generation of diversity may be very limited.

Line 587-623 (Conclusions) It would be preferable to make a clear distinction between confirmations of characteristics/observations already reported in the bibliography and new results.

Reviewer #3:

Remarks to the Author:

The authors present an improved version of the article, with a more detailed Methods section as well as a new paragraph about homeologous exchanges and chromosomal aberrations detected in various *Coffea arabica* accessions, some of which are recent. The authors propose that these rearrangements could represent an important source of genetic diversity within the species.

Overall, the comments from the reviewer were addressed in the new manuscript. However, some supplementary figures still need to be improved (some axes names/ legends, can not be read when printed: Supplementary Figures S10, S12, S13, S14, S15, S16, S17 in particular). Moreover, the Material and Methods and Supplementary Methods sections are not well balanced : most Methods are in the "Material and Methods" section, and the few added paragraphs in the "Supplementary Methods" section. A suggestion would be to move more paragraphs to "Supplementary Methods" and keep only the most important ones in the main text (or keep all paragraphs only in the main text and remove the Supplementary Methods).

The new section added (as well as re-reading of the whole manuscript) raised some additional comments/suggestions/questions to address:

1/ Line 169: TE insertions

It would be useful to have a supplementary table displaying the proportions of TE of different classes (on the whole genome, or A/B compartments) (results of EDTA).

2/ line 200 : it could be relevant to create a specific paragraph to describe the analysis of genes involved in caffeine biosynthesis. In addition, the results on expression levels are very hard to see on supplementary figures S10,S14. The "cyan" track is too small and the transcription levels can not be distinguished. Maybe displaying transcription levels with heatmaps could help visualizing the results, and could be done on one single figure for all genes at the same time.

3/ lines 409-410: "These tracts tend to extend over the same chromosomal regions across accessions as if they were preferentially retained due to selective advantages or they derived from a common ancestral event"

a- Are the functions of the genes in these tracts biased towards some specific pathways

(resistance?...)

b- Is there a correlation between the positions of shared introgressions and the positions of shared chromosomal exchanges? There seems to be hotspots of chromosomal exchanges : do they coincide with hotspots of introgressions ?

c- Is there a way to discriminate between the two hypotheses: selective advantages or common ancestral event ?

5/ Lines 474-482 : The description of the procedure to detect chromosomal aberrations in all accessions is described with a lot of details in the results section, which is not the case for other analyses: it would make sense to move the description to the Material and Methods (or Supplementary Methods) section.

6/ Line 515 : cite table S10 in addition to Figure 5 for more details on the events.

Also, on Figure 5, it would be informative to display the number of accessions where the events took place (+ the list of accessions if it is not too long).

For the events that are shared by various accessions, the authors are suggesting that they took place independently, is it correct ? Or as above, could they also correspond to shared ancestral events? It should be clarified in the text. If there are "hot spots" of homeologous exchanges in the *C. arabica* genome, these hotspots could correspond to specific functions (selective advantage, or absence of selective disadvantage) but also to specific genomic regions where HE can take place more easily (A or B compartments, Transposable Elements?), or both (or other hypotheses). Can the authors add some analyses to test these hypotheses : are the genes in the HE regions biased towards certain functions ? Are they located in specific chromosome compartments, in the vicinity of specific TE families,... ?

7/ Material and Methods, paragraph "Sequence variation and homeologous copy number variation analyses" (line 687) : the description of the method used to identify HE and reciprocal exchanges in high coverage accessions should be described with more details and maybe separated from the method used for low coverage accessions, for more clarity.

8/ Figures

Figure 1: Panel C is too small compared to A and B

Figure 2 : Panel B. Add axes on the sequence identity plot: is Chr7c (and Chr7e) on x or y ? The same goes for Figure 5, Figure S6 (and maybe others).

Figure 3: Panel B. The boxplot is not extended enough: the green boxes are almost impossible to see

Figure 5 : as mentioned previously, add axes names on the sequence identity plot (which axis corresponds to c and which axis to e). Name the events as in Table S10 for more clarity (Chr1-1, etc...). Add the number of accessions that contain the events (+names?).

Reviewer #1 (Remarks to the Author):

The genome sequence presented here is an improvement on that available but can easily be improved by the application of other methods. The value of this contribution is in the identification of the types of variation available in the coffee genome. The authors have explained their contribution well and recognized the limitations.

We thank the reviewer for the positive feedback.

Reviewer #2 (Remarks to the Author):

This revised version of the manuscript presents significant improvements over the initial version. Most of the suggestions made by the various reviewers have been taken on board, which is very much to be appreciated.

We thank the reviewer for the appreciation of the effort we made.

However, despite the efforts made, I feel that some comments have not been adequately taken into account and I am still not convinced by certain results, interpretations and conclusions.

As mentioned earlier, I have no doubts about the quality of this new genome sequence and its interest for the scientific community. The analysis of this resource carried out in the present manuscript as well as the availability and access to this new genome sequence through a dedicated website, are very much appreciated.

On the other hand, the analyses presented to study the diversification mechanisms present in *C. arabica* are less well mastered and not always correctly presented.

One of the limitations of the diversity analyses carried out is the quality of the genomic resources used (from public database). New sequencing has improved the situation, but unfortunately a large proportion of the genomic resources used are still of poor quality (and do not conform to their identification).

Pointing out errors in collections, as well as the importance of having a quality approach when duplicating collections and putting them into culture, is interesting but constitutes a difficulty for the objectives of the present study.

We think that we have addressed—with our resequencing of independent specimens and the resulting explanation in the R1 version—the limitations of using sequence information from a few critical specimens of the BioSamples SAMN10411969-71 for drawing general conclusions on the genomic constitution of the varieties that the submitting institution has associated with them.

We only partially share the discomfort expressed by the reviewer about the quality of publicly available resequencing resources. The genomic resources made available by other research groups through the NCBI short read archive and used in this manuscript conform to the standards of the repository in terms of read quality and sequence-associated

Metadata. According to the Metadata and the associated literature reports, the BioSamples subject to sequencing have been introduced from donor germplasm repositories and are maintained in the recipient germplasm repository of the institution that has generated the resequencing data, ensuring that the process of introduction and conservation was supervised by germplasm repository officers. It is true that not all sequenced specimens in the public datasets we have used might be corresponding to expectation, but it is equally true that some inaccuracies are hidden in every germplasm collection.

In the R2 version and thanks to the availability of a more complete reference genome, we have reanalyzed GBS-data of hundreds of specimens mostly held at the CATIE germplasm repository in Costa Rica (Scalabrin et al 2020 <https://www.nature.com/articles/s41598-020-61216-7>). Those data show both that the WGS panel covers comprehensively and reliably the so far available variation within the species and that no *ex-situ* repository, not even one of those that could be regarded as a gold standard for coffee genetic resources, is completely unaffected by curation inaccuracy, which likely originates from difficulties in the phenotypic identification of cultivars and landraces when the material is collected *in situ* and when entries enter repositories. Multiple entries with the same varietal name showing different genome composition and different category attribution (e.g. *bona fide* Arabica versus Timor hybrid introgression lines) represent recurrent cases in coffee, similar to those that we were confronted with when using public resequencing data.

It has to be noted that the source of GBS raw reads was genomic DNA extracted from leaf samples detached from the original stocks grown at CATIE and therefore unaffected by the technical issue of being taken from “second hand” seedlings raised from dispatched seeds of the mother plant that may have originated from unintended cross-pollination in the donor germplasm repository.

Nonetheless, before all gene banks are systematically cross-validated (which is out of the scope of this paper), we think that *bona fide* taxonomic assignment of specific entries within and among groups for population genetic analyses can ultimately be decided only upon genomic-based data mining. Therefore, trueness-to-type or adherence to expectation is, at this moment, more reliably handled *a posteriori* at the stage of data analysis rather than by relying on *a priori* information given by any plant material provider.

By treating with caution critical specimens as we did in this manuscript also thanks to the scrupulous remarks of the reviewers, we think that the inherent issue of some uncertainty in the correct identification of all genetic resources does not affect the integrity and relevance of the present study.

Another weakness of the diversity analyses is linked to the ambiguity of the objectives of the work: is it to identify the mechanisms of diversification of the *C. arabica* species since its formation (around 600,000 years ago) or to better characterize the genetic diversity of plant material in collections as part of breeding programs. For instance, the authors mention in the title “identification of chromosomal aberrations and introgressions from *Coffea canephora* as important drivers of genetic variation in the species”. While the contribution of “chromosomal aberrations” to the genetic variation identified in *C. arabica* is relatively well established,

introgressions (under natural conditions) of *Coffea canephora* into the *C. arabica* genome have never yet been identified in the course of its evolution. *Sensus stricto*, it cannot be said that introgression has been a driver of diversity in *C. arabica*. Of course, numerous artificial crosses with diploid species (including distant species such as *C. racemosa*, <https://doi.org/10.1023/A:1003427613071>) as part of breeding programs or spontaneous crosses with cultivars of diploid species (*C. canephora*, *C. liberica*) in anthropized environments (fields such as Timor Hybrid) have broadened the genetic diversity available for improving the species. For the latter, a relatively well-documented example is New Caledonia's spontaneous hybrids (<https://doi.org/10.1002/ece3.2055>; <https://doi.org/10.1139/G07-011>). These two objectives are different and must be clearly distinguished in the presentation and discussion of results.

We do not consider this as weakness or ambiguity because it is a reflection of the evolutionary history of the *C. arabica* species as a whole. Both the words “*Coffea arabica*” and the term “species” encompass spontaneous (possibly wild) populations as well as cultivated (possibly domesticated) germplasm.

Unlike in other crop species, where there is a clear separation between the wild and the cultivated germplasm in terms of their natural history and genetic diversity, in the *Coffea arabica* species, it seems to make little sense to treat separately more ancient evolution occurring between the speciation event and the beginning of human-mediated changes and agricultural exploitation and more recent evolution occurring between that moment and the present times, as ancestral genetic diversity is limited, domestication syndrome is imperceptible and among-cultivars genetic variation is very limited.

The comparison between the WGS panel and the GBS extended diversity panel that we have added to R2 supports this view.

We have added the suggested literature to accompany the explanation of the events that have contributed to broadening the genetic diversity, which are presented now more extensively in the revised Introduction Section.

In addition to specific comments, I will attempt to illustrate my points in the following paragraphs.

ABSTRACT

Lines 23-24: “a distal euchromatic one that has remained extraordinarily collinear between the two subgenomes after the split of the two species”. Difficult to understand. It would be preferable to speak of “diploid progenitor or parental genome” and not associate this observation with the allopolyploidisation step.

We have changed this wording. We are grateful for this suggestion.

Line 26-27: “low baseline intraspecific nucleotide diversity that increases to interspecific levels in a part of the Arabica-like germplasm”. Formally, this conclusion is difficult to grasp and highly debatable. For example, when a segment of the Ca subgenome (the “canephora” derived subgenome of *C. arabica*) is replaced by a C segment (from the “canephora” genotype progenitor

of the Timor hybrid) or Ea segment (the “eugenioides” derived subgenome), can this really be considered as interspecific polymorphism?

We have made this statement more precise and explicit. We thank the reviewer for pointing out that the previous wording could not be sufficiently clear to the readers.

Line 31: why “still”?

For emphasizing the fact that they are not fixed yet in the population.

INTRODUCTION

Lines 77-80: OK but only concerns genetic improvement (not the evolution of the species per se), is very partial (many other crosses have been made in breeding programs) and very biased by the publicly available whole-genome resequencing data.

OK but see above general comments on distinctions within the *C. arabica* species. References have been added in R2 to deliver the correct message that this event has had the widest impact on the diffusion of novel and divergent haplotypes into otherwise highly similar Arabica genetic backgrounds.

Lines 82-84: This remains a hypothesis (strong and not new) in the absence of results showing a direct link between phenotypic variation and chromosomal rearrangement. The term “chromosomal aberration” is not perfectly adapted. While it's well-suited to a case of aneuploidy, it's not appropriate for a simple interhomeologous segment exchange.

We have changed it to: “chromosomal aberrations of different types and exchanges between homoeologous chromosomes”

More generally, it would be interesting and totally justified to introduce bibliography on characteristics such as meiotic behavior or reproduction modalities in *C. arabica* that condition the appearance and fate of the “chromosomal aberrations” detected. Contrary to the authors' assertions, the presence of anomalies during meiosis has been known for a long time (Krug CA and Mendes AJT, 1940). Meiosis control and the fate of non-balanced gametes have been little studied in *C. arabica*. Nevertheless, see <https://doi.org/10.1093/jhered/91.1.81> and <https://doi.org/10.1139/g04-048>. With regard to the possibilities of intergenomic recombination, I have identified <https://doi.org/10.1007/s001220100747>.

We thank the reviewer for these suggestions. In the Introduction section of R2, we limited the range of bibliographic citations to those reports that focused on *C. arabica* or included *C. arabica*. Cytological evidence on meiotic behavior in *C. arabica* and flow cytometry-based DNA content variation in *C. arabica* are now introduced and discussed.

Lines 87-89: should be linked to the poor management of germplasm collections and variety distribution. The expression “Arabica-like germplasm” is a source of confusion. It must be clearly defined or avoided.

It is now defined upon first use in the text of the R2 version.

RESULTS AND DISCUSSION

Lines 200-243 (analyses of a group of genes involved in the caffeine biosynthetic) These aspects, as well mentioned in the manuscript, have already been the subject of publications. Authors should confine themselves to divergent/complementary points.

Suggestions about the relevance to be given to this information are conflictual between Reviewer 1 and Reviewer 3. See comments and replies below.

Lines 310-312 (“a reciprocal and symmetrical exchange, possibly originating from a mitotic crossing-over between homoeologous chromosomes that occurred in an interspecific hybrid diploid cell lineage before the polyploidization event”). Clarifications are needed. What are the arguments for a mitotic origin?

The reason to invoke a mitotic origin is due to the reciprocal nature of the exchange down to the single nucleotide level and the fact that we see the two reciprocal products in the same individuals. Any meiotic origin would require either that exactly the same recombination event has occurred independently in two gametes that have then given origin to the zygote or that the two pollen grains containing the reciprocal products have given origin to two plants that later on have crossed with one another. Both these events have an exceedingly low probability to occur. We think that the most likely and parsimonious explanation to account for what we observed is that a reciprocal exchange occurred mitotically in the diploid F1 hybrid that was generated by a cross btw. *C. canephora* and *C. eugenioides* and that then this event got fixed in homozygous condition when the diploid individual underwent polyploidization to become tetraploid.

In a diploid context (not linked to polyploidy), it would be preferable to speak of orthologous chromosomes. It should also be borne in mind that sequenced accessions of diploid species (*C. canephora*, *C. eugenioides*) are not the genotypes at the origin of the allopolyploidization event.

The term orthologous is not commonly used in reference to chromosomes. In the context of a diploid individual the term to be used would be homologous chromosomes but the fact is that the two chromosomes that were exchanged were not from the same species but from different parental species and thus could not be defined homologous but would be defined homoeologous even in a diploid individual. We however made some changes in the text to avoid the use of the term homoeologous in this context. We are aware of the fact that the genotypes involved in the polyploidization event do not correspond to any of the currently existing individuals and we recognized that with the use of the term present-day.

Figure 3B and corresponding text: I'm not comfortable with the way accessions are named. Knowing that a number of them aren't what they should be, a more neutral code would be better. For example, the current presentation tends to suggest that cultivars such as SL34 or Kent are introgressed when in fact they are not.

In the R2 version, we have used asterisks in the graphs of Figure 3B to mark Kent and SL34 samples (we named them anywhere in the paper in accordance with their SRA metadata) and we have pointed out in the figure legend that the sequenced specimens for Kent and SL34 correspond to BioSamples SAMN10411969 and SAMN10411970. The BioSample ID is too long to fit this panel size. However, we think that it is now mentioned multiple times across the manuscript and its figures that sequenced specimens, independently of the plant material provider and of the submitter of the sequencing data, are treated as such and cultivar names are tentatively associated with them as in their metadata. Supplementary Table S8 was updated in the R2 version with all associations between sequenced specimens and their BioSample IDs.

Having a univocal association between one single DNA profile and one cultivar name seems to be a critical issue in coffee beyond the cases that we easily spotted such as Kent and SL34 due to the unexpected introgression signature in one sequenced specimen. As multiple BioSamples from the same gene bank that go with the same cultivar name (e.g. Geisha, Laurina, Marsellesa from CATIE) are associated in SRA with raw reads batches that generate different DNA profiles and in some cases even different assignment either to *C. arabica* or to introgression lines, we think that this is a global issue that the coffee scientific community will need to address in the future with a systematic cross-validation procedure within and among gene banks using WGS.

Figure 3C and corresponding text: Since the measurement of nucleotide diversity for *C. eugenioides* is based on a single accession, it would be preferable not to present it. It might also be worth mentioning that *C. arabica* has a “fixed heterozygosity” (linked to its allopolyploid state) which represents significant nucleotide diversity.

Nei’s nucleotide diversity, expressed by π , takes into account the number of chromosomes in the population sample. π is therefore estimated even with a single accession, provided that the individual is heterozygous. Otherwise $\pi = 0$, which is not the case shown in Figure 3C. Estimates of nucleotide diversity are much less affected by a small sample size than other estimates of diversity.

Homoeologous SNPs, if the reviewer meant those sites with the use of the terms “fixed heterozygosity”, do not contribute to nucleotide diversity as it is intended in population genetics because they correspond to diversity within individuals but not within a population (they are usually fixed in all individuals). See for example Akhunov et al 2010 (Nucleotide diversity maps reveal variation in diversity among wheat genomes and chromosomes, DOI 10.1186/1471-2164-11-702) who stated: “A genome-wide assessment of nucleotide diversity in a polyploid species must minimize the inclusion of homoeologous sequences into diversity estimates”.

Line 510 (“synthetic hexaploid wheat plants present an even higher frequency of aneuploidy. The identification of aneuploid *C. arabica* accessions is therefore not unexpected”): More than a comparison with bread wheat, the identification of aneuploids was expected on the basis of our knowledge of the meiotic behavior of *C. arabica* (see previous comment above). An aneuploid *C.*

arabica plant (even dihaploid) can survive, but it is not possible to maintain it by sexual reproduction. Consequently, even if the phenomenon is frequent, its contribution to the generation of diversity may be very limited.

Aneuploidy is not necessarily maintained by sexual reproduction, but it can be because $n+1$ gametes can be transmitted from trisomic lines at a variable but not null rate. Aneuploidy can also be maintained by vegetative propagation. The findings of Ortega-Ortega et al (doi.org/10.21273/HORTSCI13916-19) that different cultivars in *C. arabica* and phenotypically diverse somatic mutants within certain cultivars show contraction or expansion in DNA content compatible with the gain or loss of one or more chromosome copies provides support to the hypothesis that both monosomy and trisomy may contribute to the generation of diversity in *C. arabica*.

Line 587-623 (Conclusions) It would be preferable to make a clear distinction between confirmations of characteristics/observations already reported in the bibliography and new results.

We have underlined in the R2 version the characteristics/observations that confirm previous reports by adding the words “as previously reported” and citing the literature reference in order to distinguish them from novel findings.

Reviewer #3 (Remarks to the Author):

The authors present an improved version of the article, with a more detailed Methods section as well as a new paragraph about homeologous exchanges and chromosomal aberrations detected in various *Coffea arabica* accessions, some of which are recent. The authors propose that these rearrangements could represent an important source of genetic diversity within the species.

Overall, the comments from the reviewer were addressed in the new manuscript. However, some supplementary figures still need to be improved (some axes names/ legends, can not be read when printed: Supplementary Figures S10, S12, S13, S14, S15, S16, S17 in particular).

We have improved the legends and axes names where suggested.

Moreover, the Material and Methods and Supplementary Methods sections are not well balanced : most Methods are in the “Material and Methods” section, and the few added paragraphs in the “Supplementary Methods” section. A suggestion would be to move more paragraphs to “Supplementary Methods” and keep only the most important ones in the main text (or keep all paragraphs only in the main text and remove the Supplementary Methods).

We agree that there was little balance between main and supplementary text in the Material and Methods section. We tried to make it more balanced in R2 by relocating paragraphs as suggested.

The new section added (as well as re-reading of the whole manuscript) raised some additional

comments/suggestions/questions to address:

1/ Line 169: TE insertions

It would be useful to have a supplementary table displaying the proportions of TE of different classes (on the whole genome, or A/B compartments) (results of EDTA).

We have added this Supplementary Table (S5 in the numbering of the R2 version).

2/ line 200 : it could be relevant to create a specific paragraph to describe the analysis of genes involved in caffeine biosynthesis. In addition, the results on expression levels are very hard to see on supplementary figures S10,S14. The “cyan” track is too small and the transcription levels can not be distinguished. Maybe displaying transcription levels with heatmaps could help visualizing the results, and could be done on one single figure for all genes at the same time.

Suggestions about the relevance to be given to this information are conflictual between Reviewer 2 and Reviewer 3. As a compromise, we have decided not to give it more relevance and to maintain it in the paragraph where it was embedded, as our discussion on this issue is mainly focused on the comparative structural analysis of the loci containing those genes between the two subgenomes and gene expression analysis is given as an accompanying information.

The cyan track is intended to provide a base-by-base coverage plot that shows aligned RNA-Seq reads supporting the gene model predictions. We agree that differences in transcription levels could not be visually appreciated from the same graphs, but we have reported in the original submission as well as in the previous R1 version the normalized levels of gene expression in graphical format (bar plots indicating TPM) in Supplementary Figure S11 (one single figure for all genes at the same time). We added a note in the legends of Supplementary Figures S10 and S14 to redirect readers who wish to visualize this information to Supplementary Figure S11.

3/ lines 409-410: “These tracts tend to extend over the same chromosomal regions across accessions as if they were preferentially retained due to selective advantages or they derived from a common ancestral event”

a- Are the functions of the genes in these tracts biased towards some specific pathways (resistance?...)

We have generated the requested information focusing within each chromosomal region around the peak of introgressed haplotype frequency (Supplementary Figure S29) and added the results of GO enrichment analysis as a new Supplementary Table (S9 in the R2 version). However, we think that the linkage drag around an hypothetical selectively advantageous locus within each introgressed chromosome segment (see Figure 3C), which extends in many cases for several millions Megabases, is a major impediment against the detection of statistically significant enrichments within or around the putative target of selection, because the target gene in each introgressed segment is outnumbered by genes

that are simply in LD and are likely to belong to unrelated functional categories. Nonetheless, in the cases where we could restrict the region to a sharp peak of high introgressed haplotype frequency, we detected significant enrichment in genes belonging to expected biological processes and in the expected direction.

b- Is there a correlation between the positions of shared introgressions and the positions of shared chromosomal exchanges? There seems to be hotspots of chromosomal exchanges: do they coincide with hotspots of introgressions ?

We generated a new Supplementary Figure S32 that compares the genomic location of the breakpoints of homologous recombination between *C. arabica* haplotypes and HDT-derived haplotypes that increase the levels of diversity in the canephora subgenome in *C. arabica* × *C. canephora* introgression lines and the sites of homoeologous exchanges in *C. arabica*. While both set of events tend to occur in regions that are enriched in genes, depleted in TEs and more collinear between homoeologs than the average of the chromosome in which they have occurred, there is no overlapping at a finer scale (Supplementary Tables S14-S15 and Source Data: Suppl Fig. S28-S32, S36, S42A), as if the two phenomena are stochastic or driven by independent forces.

c- Is there a way to discriminate between the two hypotheses: selective advantages or common ancestral event ?

We are not confident in drawing more general conclusions than those presented for the specific points a- and b-.

5/ Lines 474-482 : The description of the procedure to detect chromosomal aberrations in all accessions is described with a lot of details in the results section, which is not the case for other analyses: it would make sense to move the description to the Material and Methods (or Supplementary Methods) section.

We agree that this description was disproportionately long and detailed in the Result section compared to the rest of the analyses. We moved it to Supplementary Methods.

6/ Line 515 : cite table S10 in addition to Figure 5 for more details on the events.

We added the reference to the Supplementary Table.

Also, on Figure 5, it would be informative to display the number of accessions where the events took place (+ the list of accessions if it is not too long).

We added the information about the number of accessions in which the event was detected. The List of all accessions is too long to fit into Figure 5.

For the events that are shared by various accessions, the authors are suggesting that they took

place independently, is it correct ? Or as above, could they also correspond to shared ancestral events? It should be clarified in the text.

We apologize for this ambiguity. For the sake of parsimony, we treated them as they were shared by descent as we could not demonstrate they took placed independently.

“We tried to cluster events with similar chromosomal coordinates to indicate shared events among accessions (Figure 5, Supplementary Table S10 and Supplementary Figure S33), even though the low resolution of the analysis due to the large window size used as a consequence of the low read coverage does not warrant their identity by state.”

The ambiguity resides in the last word of our sentence, which we recognize was imprecise in R1. “Identity by state” has now been replaced by “identity by descent” in the R2 version.

If there are “hot spots” of homeologous exchanges in the *C arabica* genome, these hotspots could correspond to specific functions (selective advantage, or absence of selective disadvantage) but also to specific genomic regions where HE can take place more easily (A or B compartments, Transposable Elements?), or both (or other hypotheses). Can the authors add some analyses to test these hypotheses : are the genes in the HE regions biased towards certain functions ? Are they located in specific chromosome compartments, in the vicinity of specific TE families,... ?

We have generated the requested information and added it as a new Supplementary Table reporting gene and TE content as well as the relative extent of A/B compartments and the level of collinearity within the exchanged regions compared to the average of the chromosome in which each HE has occurred. We also run GO enrichment analysis on the set of genes located in the exchanged regions without finding any significant enrichment or depletion for functional categories. We did not discuss this last point as we did not have any expectation that gene function could be a driver of for HE.

7/ Material and Methods, paragraph “Sequence variation and homeologous copy number variation analyses” (line 687) : the description of the method used to identify HE and reciprocal exchanges in high coverage accessions should be described with more details and maybe separated from the method used for low coverage accessions, for more clarity.

The method and parameters that we used to identify HE and reciprocal exchanges are always identical for all accessions, no matter the genome-wide coverage of the sample. With high coverage accessions, the methods and the parameters (fixed factors) were run repeatedly using reduced subsets of input reads in order to simulate increasingly lower genome-wide coverages (variable factor) as those encountered in most of the samples obtained from public dataset. As this part has now been moved to Supplementary Information, we explained this procedure more extensively.

8/ Figures

Figure 1: Panel C is too small compared to A and B

The size of the chromosomal diagrams are identical among panels A, B and C. The diagram of each chromosome in panel C has been divided into three sections to respond to the reviewer 3 previous request to include here also the information of A/B compartments that is then shown in greater graphical detail in Supplementary Figure S3. We think that the heatmaps in blue and red for exon and TE densities, respectively, in panel C are graphically quite clear at the current size, for the level of resolution that this figure is intended to convey (and indeed the Reviewer did not complain about this in the previous round of reviewing). We agree with the remarks of the Reviewer that the gray versus white colors we used in R1 to indicate the A-type and B-type windows has lower chromatic contrast than the other sectors and the information it should convey is visually less evident at this size. In R2, we tried to maintain the size of the chromosomal diagrams unchanged (for consistency with other panels) and to make the critical sector in panel C more evident using a more vivid color.

Figure 2 : Panel B. Add axes on the sequence identity plot: is Chr7c (and Chr7e) on x or y ? The same goes for Figure 5, Figure S6 (and maybe others).

We have added the axes in the identity plot section of panel B of Figure 2. Axis titles indicating that Chr7c is along x-axis and y-axis Chr7e is along were present in the original and in R1 versions.

As for Figure 5, x- and y-axis are both shown outermost with respect to both the identity plots and the bar plots, which refer to the same scale of chromosomal coordinates. In the R2 version, we have mentioned this in the figure legend. We also have added the other axis title of the bar plot (Fraction) that was missing from previous versions.

As for Supplementary Figure S6, in the R2 version we have mentioned in the legend that the axis indicating the chromosomal coordinates in Mbp of each homoeolog refers to both the bar plot and the sequence identity plot.

Figure 3: Panel B. The boxplot is not extended enough: the green boxes are almost impossible to see

This is indeed the information the green boxes are intended to convey: a consistently low number of SNPs in all windows that represent the *C. arabica* genetic background within the genomes of introgression lines. The fact that, due to this low variation, the olive green body of the box is not visible because the black lines for drawing the borders of the box are thicker than the vertical size of the box does not introduce ambiguity, as it is defined in the Figure Legend that olive green boxes are those that are plotted to the left of each variety while the magenta boxes (those reporting the counts in windows across the introgressed regions) are plotted to the right of each variety.

Figure 5 : as mentioned previously, add axes names on the sequence identity plot (which axis corresponds to c and which axis to e). Name the events as in Table S10 for more clarity (Chr1-1, etc...). Add the number of accessions that contain the events (+names?).

We have added in the R2 version the indication that the canephora homoeolog is shown on the x-axis and the eugenioides homoeolog is shown on the y-axis using the chromosomal coordinates axis title.

We have also added the extended full code indicating each event of homoeologous exchange as in Supplementary Table S15.

We have reported in the R2 version the number of accessions in which each HE event has been identified. The list of accessions in which each HE event has been identified doesn't fit the size of this figure and it is both reported in tabular format in Supplementary Table S15 and the same information is provided in graphical format in Supplementary Figure S41.

Reviewers' Comments:

Reviewer #2:

Remarks to the Author:

This revised version of the manuscript presents significant improvements over the previous version. The main requests have been met in one way or another. The presentation of the results is better balanced and the limitations of the study clearly presented.

I have only one very minor remark on a new paragraph introduced by the authors: Based on the work of Ortega et al. (2019), it is mentioned (lines 93-94 and lines 555-556) that "The frequency and the extent of DNA content variation that are observed among cultivars and among phenotypic mutants within cultivars could provide support for the presence of aneuploidies in *C. arabica* cultivated germplasm". While it's always difficult to rule out the possibility of an exceptional event, this new argument is hardly credible. Indeed, the existence of aneuploid cultivars is highly unlikely. As the cultivars studied are propagated by seed (following self-fertilization), an aneuploid state would be a source of heterogeneity. On the other hand, the cultivar's fertility would be affected and associated with the formation of round coffee beans (i.e. Peaberry) due to the abortion of one of the two ovules in the ovary, thus freeing space for the single developing seed. Peaberry frequency may vary between cultivars, but it is a trait that is strongly counter-selected during the breeding process. In the referenced article, the authors associate these genome size variations with the presence of *Canephora* introgressions (which the present work confirms once again).

Reviewer #3:

Remarks to the Author:

The new version of the manuscript addresses all comments previously raised by the reviewer.

REVIEWERS' COMMENTS

Reviewer #2 (Remarks to the Author):

This revised version of the manuscript presents significant improvements over the previous version. The main requests have been met in one way or another. The presentation of the results is better balanced and the limitations of the study clearly presented.

I have only one very minor remark on a new paragraph introduced by the authors: Based on the work of Ortega et al. (2019), it is mentioned (lines 93-94 and lines 555-556) that “The frequency and the extent of DNA content variation that are observed among cultivars and among phenotypic mutants within cultivars could provide support for the presence of aneuploidies in *C. arabica* cultivated germplasm”. While it's always difficult to rule out the possibility of an exceptional event, this new argument is hardly credible. Indeed, the existence of aneuploid cultivars is highly unlikely. As the cultivars studied are propagated by seed (following self-fertilization), an aneuploid state would be a source of heterogeneity. On the other hand, the cultivar's fertility would be affected and associated with the formation of round coffee beans (i.e. Peaberry) due to the abortion of one of the two ovules in the ovary, thus freeing space for the single developing seed. Peaberry frequency may vary between cultivars, but it is a trait that is strongly counter-selected during the breeding process. In the referenced article, the authors associate these genome size variations with the presence of *Canephora* introgressions (which the present work confirms once again).

While we appreciate that the existence of aneuploid cultivars is highly unlikely, and we understand that the differences observed by Ortega et al. among cultivars are more likely to be due to introgressions from *Canephora*, especially when looking at lines where introgressions are known to have taken place, we find it harder to accept that explanation for spontaneous mutants within cultivars. Ortega et al are stating that “Mechanisms underlying intraspecific and interspecific genome size variation in plants, particularly at the evolutionary level are not well understood; thus, more research is required in this regard. “. We think that aneuploidies could provide an explanation at least for some of the cases and have modified the two sentences by removing the reference to differences among cultivars and leaving the reference to differences among phenotypic mutants within cultivars.

Reviewer #3 (Remarks to the Author):

The new version of the manuscript addresses all comments previously raised by the reviewer.

We thank the reviewer for the positive feedback.